# 2*H*-Thiopyran-2-thione sulfine, a compound for converting H₂S to HSOH/H₂S₂ and increasing intracellular sulfane sulfur levels

Qi Cui [1,6], Meg Shieh [1,6], Tony W. Pan [1], Akiyuki Nishimura [2], Tetsuro Matsunaga [3], Shane S. Kelly [4], Shi Xu [1], Minkyung Jung [3], Seiryo Ogata [3], Masanobu Morita [3], Jun Yoshitake [3], Xiaoyan Chen [1], Jerome R. Robinson [1], Wei-Jun Qian [4], Motohiro Nishida [2,5], Takaaki Akaike [3] ✉ & Ming Xian [1] ✉

Reactive sulfane sulfur species such as persulfides (RSSH) and H₂S₂ are important redox regulators and closely linked to H₂S signaling. However, the study of these species is still challenging due to their instability, high reactivity, and the lack of suitable donors to produce them. Herein we report a unique compound, *2H*-thiopyran-2-thione sulfine (TTS), which can specifically convert H₂S to HSOH, and then to H₂S₂ in the presence of excess H₂S. Meanwhile, the reaction product *2H*-thiopyran-2-thione (TT) can be oxidized to reform TTS by biological oxidants. The reaction mechanism of TTS is studied experimentally and computationally. TTS can be conjugated to proteins to achieve specific delivery, and the combination of TTS and H₂S leads to highly efficient protein persulfidation. When TTS is applied in conjunction with established H₂S donors, the corresponding donors of H₂S₂ (or its equivalents) are obtained. Cell-based studies reveal that TTS can effectively increase intracellular sulfane sulfur levels and compensate for certain aspects of sulfide:quinone oxidoreductase (SQR) deficiency. These properties make TTS a conceptually new strategy for the design of donors of reactive sulfane sulfur species.

Since the discovery of hydrogen sulfide (H₂S) as a nitric oxide (NO)-like signaling molecule in recent years, research on H₂S and its related reactive sulfur species has exploded[1–3]. A hot topic in this field is to develop compounds or materials that can precisely control the release of H₂S or its related sulfur species and to explore their applications[4–8]. In this regard, hydrogen persulfide (H₂S₂), a highly reactive sulfane sulfur species, has received much attention as it is believed to have distinct regulatory functions in redox biology and is closely linked to H₂S-signaling[9–11]. Endogenous H₂S₂ is produced indirectly by enzymes like 3-mercaptopyruvate transferase (3-MST) and cysteinyl-tRNA

synthetase (CARS), via persulfides as the key intermediates. It can also be produced from H₂S via H₂S-NO cross talk or hemeprotein-catalyzed oxidation. H₂S₂ appears to be more efficient than H₂S in inducing protein persulfidation (forming P-SSH), and this is suggested to be one of the main functions of H₂S₂ in biology[12–15].

While the research on H₂S₂ is rapidly growing, a major challenge is the use or handling of the H₂S₂ source in studies. H₂S₂ is much more reactive and unstable than H₂S. Therefore, there is a pressing need to develop suitable H₂S₂ donors. Unlike H₂S, whose donors have been extensively studied with a large number of such compounds already

[1]Department of Chemistry, Brown University, Providence, RI 02912, USA. [2]Division of Cardiocirculatory Signaling, National Institute for Physiological Sciences (NIPS) and Exploratory Research Center on Life and Living Systems (ExCELLS), National Institutes of Natural Sciences, Okazaki 444-8787, Japan. [3]Department of Environmental Medicine and Molecular Toxicology, Tohoku University Graduate School of Medicine, Sendai 980-8575, Japan. [4]Biological Sciences Division, Pacific Northwest National Laboratory, Richland, WA 99352, USA. [5]Department of Physiology, Graduate School of Pharmaceutical Sciences, Kyushu University, Fukuoka 812-8582, Japan. [6]These authors contributed equally: Qi Cui, Meg Shieh. ✉e-mail: takaike@med.tohoku.ac.jp; ming_xian@brown.edu

reported[4–8], the donors of $H_2S_2$ are still very limited. Currently, most studies use $Na_2S_2$ as the equivalent of $H_2S_2$, but $Na_2S_2$ is considered to be an uncontrollable and instant $H_2S_2$ donor. A few synthetic $H_2S_2$ donors have been reported, and their structures are shown in Fig. 1-A[16–20]. These donors rely on protected disulfide (-S-S-) structures, which have inherent limitations: 1) disulfides are sensitive to disulfide-exchange reactions with cellular thiols (Cys or GSH) and would change their identity (e.g. not form $H_2S_2$), and 2) acyl disulfides (as shown in I and BW-HP, Fig. 1-A) are highly reactive toward cellular nucleophiles (-SH, -NH₂, etc.), which can diminish their efficiency. In theory, $H_2S_2$ can be produced from in situ reactions of $H_2S$ with oxidants such as $H_2O_2$ and $HClO_4$[21]. However, this is not a suitable strategy for $H_2S_2$ delivery as the presence of oxidants can compromise the biological systems under investigation. In addition, the oxidation can hardly be controlled, so over-oxidation could occur to produce the stable and much less reactive adduct, elemental sulfur ($S_8$). The specificity of oxidation is also a problem as many biomolecules (such as proteins) can be modified by oxidation conditions. Given the aforementioned considerations, exploring novel strategies in the design of $H_2S_2$ donors has been a top priority in our lab. Herein, we report a unique booster system based on *2H*-thiopyran-2-thione sulfine (TTS), which can effectively and specifically convert $H_2S$ to HSOH, and then to $H_2S_2$ via non-oxidation and non-disulfide conditions. The reaction mechanism has been studied, and its applications in protein and cell models are

evaluated. Interestingly, the reaction product *2H*-thiopyran-2-thione (TT) can be converted back to TTS by biologically relevant oxidants, highlighting this as a potentially regenerable system. Furthermore, TTS can be applied in conjunction with $H_2S$-donor systems, essentially turning the $H_2S$ donor system into an HSOH/$H_2S_2$-donor system.

## Results

### Design and preparation of the sulfine-based booster TTS

In our search of possible new chemistry for $H_2S_2$ generation, we identified sulfines (thiocarbonyl oxides) as having great potential. Sulfines are oxidative intermediates of thiocarbonyl compounds[22–25]. Aliphatic sulfines were found to be normally unstable and could decompose to form the corresponding carbonyl and thiocarbonyl compounds with half-lives of 1–4 days (Fig. 1B). When UV light was introduced, atomic sulfur could release[23]. Nucleophiles such as amines could react with sulfines to form imines and HSOH. It is worth noting that HSOH is the smallest sulfenic acid, which can be expected to possess $H_2S_2$-like reactivity (such as inducing persulfidation). However, the chemistry of HSOH has not been well-studied due to its instability. To the best of our knowledge, there is no HSOH donor reported thus far. Based on these analyses, sulfines may serve as unique HSOH (or sulfane sulfur) donors. However, the poor stability of common sulfines would limit their applications. We envisioned that if the sulfine moiety was introduced to an aromatic heterocyclic system, its stability would

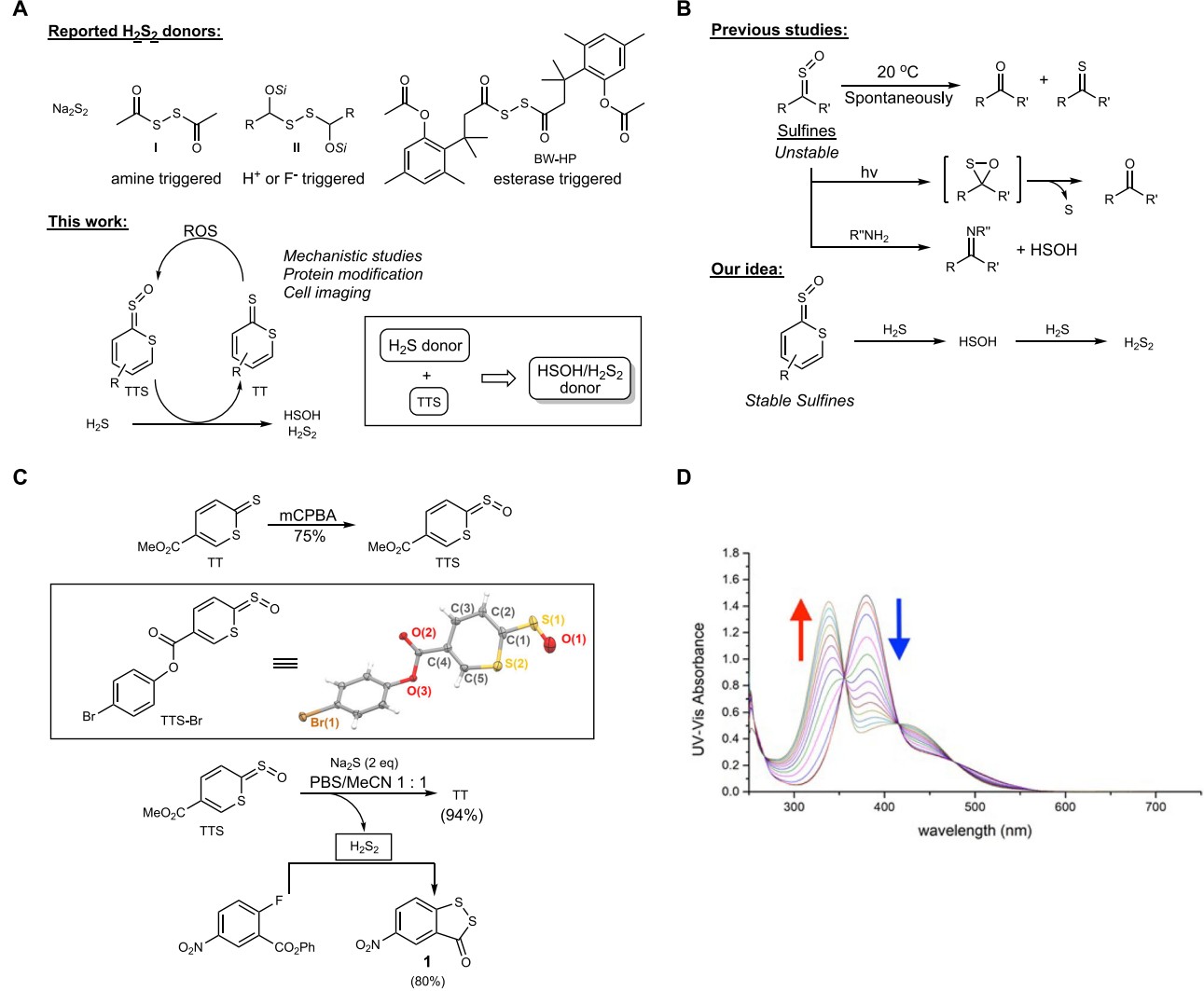

**Fig. 1 | Summary of the idea and chemistry of TTS. A** Reported $H_2S_2$ donors and the proposed HSOH/$H_2S_2$ releasing system. **B** The idea of using sulfines to produce HSOH/$H_2S_2$. **C** Preparation and reaction of TTS. **D** UV-Vis absorbance spectra changes of the reaction between TTS (100 μM) and $H_2S$ (200 μM) over a 1 min interval.

be significantly enhanced. That could prevent its reaction with less reactive cellular nucleophiles (-OH, -NH$_2$, etc.) and only allow the most reactive nucleophiles (such as H$_2$S) to react. More importantly, once HSOH is generated, it could further react with H$_2$S to form H$_2$S$_2$, thereby creating a system for H$_2$S$_2$ generation.

Our recent work showed that thiocarbonyl-containing heterocycles such as *2H*-thiopyran-2-thiones (TT) could serve as CS$_2$ donors upon bio-orthogonal cycloaddition with strained alkynes[26]. Thus, these heterocycles may be appropriate substrates to use for sulfine preparation and allow us to test our hypothesis. As shown in Fig. 1C, we found that the sulfine derivative (TTS) of *2H*-thiopyran-2-thiones could be readily prepared from TT upon the treatment of *m*-CPBA. The structure of TTS was unambiguously determined by single-crystal X-ray diffraction studies of the analog, TTS-Br (Fig. 1C and Supplementary Figs. 1 and 2). Both the heterocyclic core and sulfine fragment were effectively coplanar (∠S(2)–C(1)–S(1)–O(1): 0.1(7)°), where the S−O bond was oriented towards the endocyclic sulfur. The S−O bond distance in TTS-Br (1.381(10) Å) supported significant double bond character and was shorter than that of aryl sulfines (1.427(4)−1.477(2) Å)[27,28] and close to those found in interstitial SO$_2$ (-1.380 Å)[29–31].

## Reactions of TTS

Next, we tested TTS's reaction with H$_2$S. In a mixed solvent system with PBS buffer (pH 7.4) and acetonitrile (1:1), the reaction between TTS and H$_2$S (using Na$_2$S as the equivalent) produced TT in an almost quantitative yield (94%). The formation of H$_2$S$_2$ in this reaction was also demonstrated by trapping H$_2$S$_2$ with phenyl 2-fluoro-5-nitrobenzoate[32], which gave the desired product 5-nitro-3*H*-1,2-benzodithiol-3-one **1** in an 80% isolated yield. Since HSOH was expected to be the key intermediate in the reaction, we attempted to trap HSOH by chemicals such as dimedone. However, we were unable to obtain the

desired HSOH-trapped product. This might be attributed to the much faster reaction between HSOH and H$_2$S than the trapping reaction. Nevertheless, the mechanistic analyses (*vide infra*) still support the formation of HSOH in this process.

It is worth noting that the reaction between TTS and H$_2$S was a clean and fast process. TTS and TT possess different UV-Vis absorbance spectra at 380 and 339 nm, respectively, so the reaction could be easily monitored by UV-Vis spectrophotometry (Fig. 1D). By measuring time-dependent spectra changes, the second-order rate constant, $k_2$, was calculated to be 2.7 M$^{-1}$s$^{-1}$, indicating that this was a fast reaction comparable to some click reactions[33].

The reaction between TTS and H$_2$S is unique as it produces not only H$_2$S$_2$, but also TT, the precursor of TTS. As such, this molecule might be regenerated in biological systems upon reaction with oxidants. To test this idea, we first treated TT with H$_2$O$_2$, the most prevalent oxidant in biology. As shown in Fig. 2A, the reaction was clean and reached completion in -24 h. The progress of the reaction could also be monitored by UV-Vis spectrophotometry, where the reaction rate constant ($k_2$) was measured to be 0.1 M$^{-1}$s$^{-1}$. We further measured the oxidation of TT by several other oxidants, including *t*-BuOOH, AcOOH, and OCl$^-$, and the results were summarized in Fig. 2B. *t*-BuOOH showed weaker reactivity than H$_2$O$_2$ while AcOOH and OCl$^-$ were efficient and caused rapid oxidation. These results demonstrated that the TTS-based booster system could be regenerated in the presence of appropriate oxidants.

The TTS/H$_2$S reaction mechanism was proposed in Fig. 2C, which was also studied by DFT (M06-2X) calculations. The reaction was initiated by the nucleophilic addition of HS$^-$ to the thiocarbonyl carbon of TTS via TS1 to give intermediate INT1. This rate-determining step had an activation Gibbs free energy of 15.2 kcal/mol. Then, the [1,4]-H shift on INT1 occurred to give thermodynamically

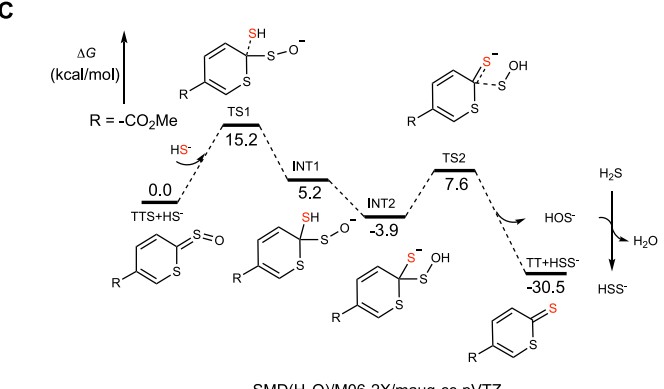

| ROS | $k_2$ |
|---|---|
| H$_2$O$_2$ | 0.1 M$^{-1}$ s$^{-1}$ |
| $^t$BuOOH | 1.0 x 10$^{-2}$ M$^{-1}$ s$^{-1}$ |
| AcOOH | 13.9 M$^{-1}$ s$^{-1}$ |
| OCl$^-$ | > 1.0 x 10$^3$ M$^{-1}$ s$^{-1}$ $^a$ |

**Fig. 2 | Mechanistic studies of TTS reactions. A** NMR spectra changes from the oxidation of TT to form TTS. **B** Summary of the TT oxidation rate constants $k_2$. TT (100 or 10 μM) and ROS (1 mM) were reacted in PBS buffer, pH = 7.4 with DMSO (1%) at rt. The kinetics were determined by UV-Vis (λ = 339 nm) based on the decrease of TT. $^a$The rate constant was estimated because the reaction was too fast. Source data are provided as a Source Data file. **C** Theoretical studies of the reaction between TTS and H$_2$S. Gibbs free energies of all the intermediates and transition states were computed at the SMD(H$_2$O)/M06-2X/maug-cc-pVTZ level. **D** $^{34}$S-isotope labeling experiments.

favored INT2. A sulfur(0) species HOS⁻ was released next via TS2 ($\Delta G^{\ddagger} = 11.5$ kcal/mol), which could react with another molecule of $H_2S$ to give $HSS^-$, $H_2O$, and TT.

To provide additional experimental support for the proposed mechanism, we carried out a sulfur isotope labeling experiment (Fig. 2D). TTS was treated with ³⁴S-labeled $Na_2S$ in PBS buffer (pH 7.4). As expected, ³⁴S-labeled product TT-³⁴S was obtained and characterized by MS analysis (Supplementary Fig. 3). It should be noted that TT-³⁴S could be converted into TTS-³⁴S (Supplementary Fig. 4) upon oxidation. Therefore, TTS-³⁴S could be a useful reagent for the in-situ generation of ³⁴S-labeled HSOH/$H_2S_2$, and we believe that this could have some useful applications in understanding the biological functions of these species. As a proof of concept, we used 2-fluoro-5-nitrobenzoate to trap ³⁴S-labeled $H_2S_2$. As expected, ³⁴S-labeled product **1**-³⁴S was obtained, albeit with the possibility of having two isomers (the HRMS spectrum of **1**-³⁴S is shown in Supplementary Fig. 5).

### Validation of $H_2S_2$ formation from TTS by fluorescence studies

To further evaluate the specificity of TTS in converting $H_2S$ to $H_2S_2$ under our conditions (e.g. w/ excess $H_2S$), we used SSP4 to monitor the formation of $H_2S_2$ in this reaction and compared the results with a series of controls. SSP4 is a well-established fluorescent sensor for sulfane sulfurs including $H_2S_2$[34]. As shown in Fig. 3I-A, only the co-existence of $H_2S$ and TTS yielded significant fluorescence response. We did not observe any fluorescence in the cases of TTS/TT only, TTS+Cys, TT + $H_2S$, and TT+Cys. Interference experiments with Cys or GSH were also tested (Fig. 3I-B). The presence of thiols led to decreased but still significant fluorescence. This was expected as thiols would compete with SSP4 to react with $H_2S_2$, thereby causing decreased signals. Taken together, these results demonstrated that TTS could effectively convert $H_2S$ to $H_2S_2$. The results also showed that the byproduct TT would not cause unwanted reactions with Cys or $H_2S$.

To demonstrate that the reaction between TTS and $H_2S$ could proceed under cellular environments, we carried out live cell imaging studies. As shown in Fig. 3II, HeLa cells were treated with TTS, TTS + $H_2S$, or TT for 30 min before the fluorescent sensor SSP4 was applied to the cells to monitor the formation of $H_2S_2$. As expected, we only observed significant green fluorescence in cells with both TTS and $H_2S$ (Fig. 3II-D). The other two treatments (TTS only or TT only) did not show obvious fluorescence. $Na_2S_2$ was used as the positive control in this study (Fig. 3II-F). Other control studies including TT + $H_2S$, $H_2S$ only, and SSP4 only were also performed, and we did not observe obvious fluorescence in these samples (Fig. 3II-G-I). These results demonstrated that TTS could effectively convert $H_2S$ to $H_2S_2$ in complex biological systems. It should be noted that the actual cellular reactions might be more complex due to the formation of HSOH in this process. However, HSOH should possess $H_2S_2$-like reactivity and rapidly react with cellular thiols to form persulfides (RSSH), which would yield the same result as that from $H_2S_2$. All these reactive sulfane sulfur species ($H_2S_2$, HSOH, RSSH) can be detected by SSP4 and can be considered as $H_2S_2$ equivalents here. Therefore, in the following biological and biologically-relevant studies, mentions of $H_2S_2$ should be understood as those of $H_2S_2$ and/or its equivalents in those systems.

To determine whether TTS could also convert endogenous $H_2S$ to $H_2S_2$ in biological systems, we decided to induce hypoxia in cells because hypoxic environments are known to elevate $H_2S$ levels[35–37]. Neonatal rat cardiomyocytes (NRCMs) were used in this study. Hypoxia-induced elevated $H_2S$ in NRCMs was first confirmed by SF7-AM, an $H_2S$-specific fluorescent probe[38] (Supplementary Fig. 8). Next, NRCMs were treated with TTS or TT (10 μM) under hypoxic conditions and then incubated with SSP4 (5 μM) for imaging (Fig. 4). For comparison, the same experiments were also performed with cells under normoxia. As expected, strong fluorescence from SSP4 was observed in the cells treated with TTS with significantly lower fluorescence in TT-treated or untreated cells, or in cells under normoxia. These results demonstrated the applicability of TTS as a booster for endogenous $H_2S_2$ (or its equivalents) in cellular systems.

### Quantitative determination of intracellular sulfane sulfur levels with TTS treatment

We further conducted sulfur metabolome analyzes using beta-(4-hydroxyphenyl)ethyl iodoacetamide (HPE-IAM) to quantitatively measure the levels of sulfane sulfurs levels in TTS-treated cells. Mouse embryonic fibroblasts (MEFs) were treated with 100 μM TTS or 50 μM $Na_2S_2$ for 1 h at 37 °C, followed by washing and application to the sulfur metabolome. The results shown in Fig. 5 revealed that TTS alone was able to preferentially increase the level of certain persulfides such as GSSH, $H_2S_2$ and $H_2S_3$ in MEF cells. $Na_2S_2$ also significantly increased the levels of these persulfides. These findings thus suggest that TTS indeed promoted sulfur catenation, leading to the production of reactive sulfane sulfurs such as persulfides and polysulfides, even in the cellular context.

### Preparation and validation of protein-bound TTS

The reaction between TTS and $H_2S$ can be considered a bio-orthogonal reaction, so we expected that TTS could be delivered to certain cellular locations or conjugated with certain biomolecules to achieve specific/localized $H_2S$ to $H_2S_2$ transformation. To test this idea, we prepared a TTS-conjugated p-nitrophenol carbonate **2** and treated it with a model protein −lysozyme C. As shown in Fig. 6, the TTS moiety was successfully linked to the protein. When the modified protein (Lyso-TTS) was then treated with $H_2S$, the desired product Lyso-TT was obtained (as demonstrated by MS). The formation of $H_2S_2$ in this reaction was also confirmed (by the SSP4 assay). These results further demonstrated the stability and applicability of TTS in biological systems. For example, protein-bound TTS can be engineered for controlled $H_2S_2$ delivery or controlled induction of protein modifications via protein-protein interactions.

### TTS-induced protein persulfidation

One of the primary functions of $H_2S_2$ is to induce protein S-persulfidation. To achieve this, researchers usually treat proteins with $Na_2S_2$ directly to obtain the persulfidated proteins. However, it is known that the use of continuous and lower concentrated oxidants is often more effective than that of the bolus addition of the oxidants to cause biological responses[39–41]. We thus wondered if the TTS + $H_2S$ system would be more efficient at persulfidating proteins than $Na_2S_2$. GAPDH was used as the protein model, and its persulfidation was evaluated by SSP4 using a protocol we recently established[34,39]. In this study, the reduced GAPDH (100 μM) was individually treated with 10 eq. of TTS+$Na_2S$ or $Na_2S_2$, as well as control reagents (TTS or $Na_2S$ only). The protein samples were then purified using Zeba Spin Desalting Columns (7 K MWCO), diluted to 10 μM, and treated with SSP4 (5 μM). Fluorescence signals were recorded to reflect persulfidation levels. As shown in Fig. 7A, control samples did not give any notable fluorescence. Samples treated with TTS+$Na_2S$ or $Na_2S_2$ yielded strong fluorescence while statistical analyses revealed that the combination of TTS and $Na_2S$ gave significantly higher signals than that of $Na_2S_2$. To demonstrate that the observed fluorescence signals were not due to the insufficient removal of $Na_2S_2$ in the desalting step, we carried out control studies using -SH blocked GAPDH, small molecules only samples, and cysteine-mutated recombinant human GAPDH under the same conditions. These samples (shown in Supplementary Fig. 9) did not give any fluorescence. To provide further evidence of persulfidation in the GAPDH system, TTS+$Na_2S$-treated GAPDH was alkylated with HPE-IAM to stabilize the protein persulfides. The protein was then digested with trypsin, and the peptides were subjected to LC-MS/

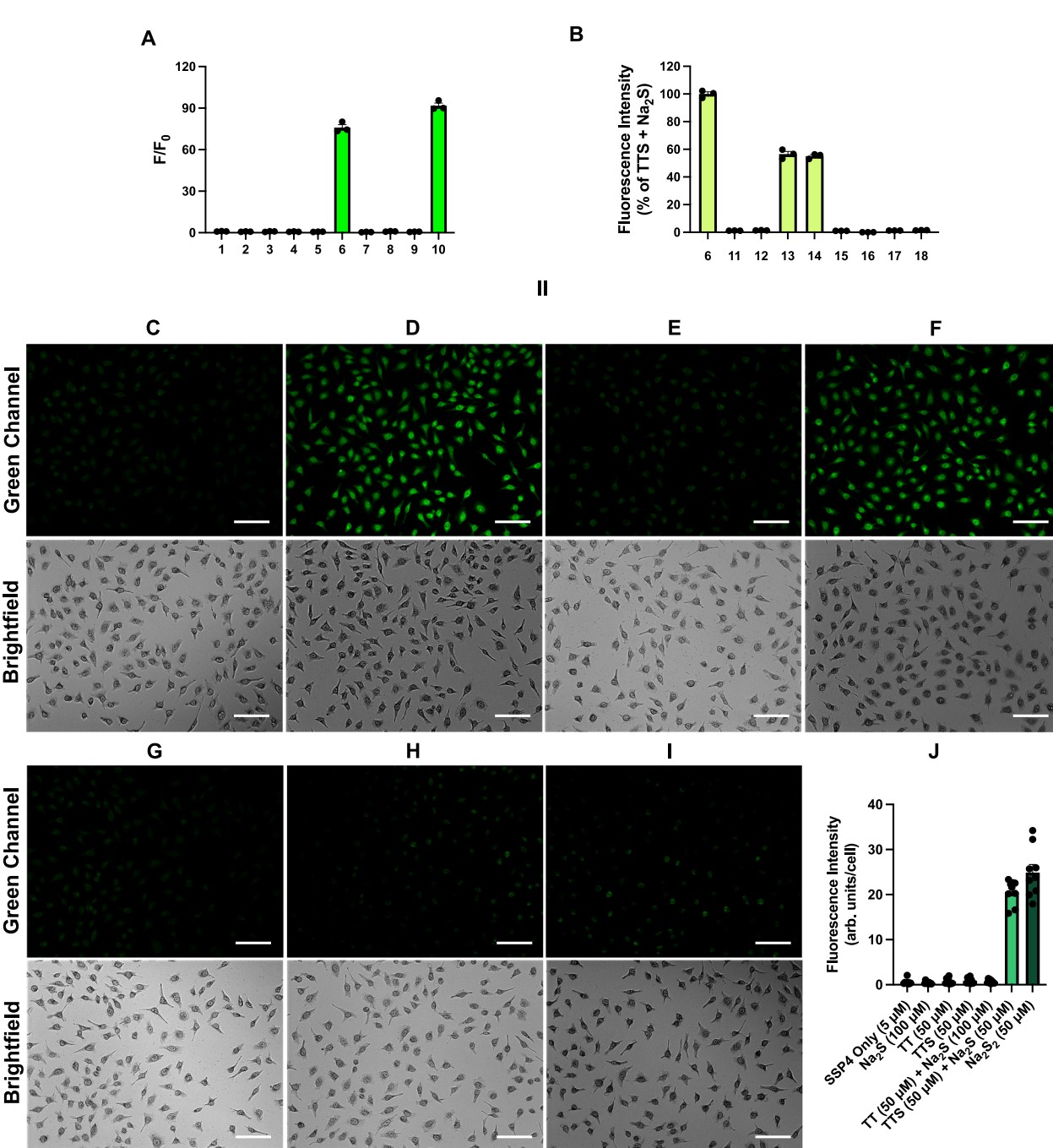

**Fig. 3 | Fluorescence-based studies of H$_2$S$_2$ formation from TTS. I)**
**A** Fluorescence enhancements of SSP4 (5 μM) in the presence of: (1) SSP4 only; (2) 25 μM TTS; (3) 50 μM Na$_2$S; (4) 100 μM Cys; (5) 25 μM TT; (6) 25 μM TTS + 50 μM Na$_2$S; (7) 25 μM TTS + 100 μM Cys; (8) 25 μM TT + 50 μM Na$_2$S; (9) 25 μM TT + 100 μM Cys; (10) 25 μM Na$_2$S. **B** Percent fluorescence of (11) 25 μM TTS + 200 μM Cys; (12) 25 μM TTS + 1 mM GSH; (13) 25 μM TTS + 50 μM Na$_2$S + 200 μM Cys; (14) 25 μM TTS + 50 μM Na$_2$S + 1 mM GSH; (15) 25 μM TT + 200 μM Cys; (16) 25 μM TT + 1 mM GSH; (17) 25 μM TT + 50 μM Na$_2$S + 200 μM Cys; (18) 25 μM TT + 50 μM Na$_2$S + 1 mM GSH compared to (6). Results are expressed as mean ± SEM ($n$ = 3 distinct samples). **II)** Representative fluorescence images of the reaction between TTS and H$_2$S in HeLa cells. Cells were first treated with **C** TTS only (50 μM), **D** TTS (50 μM) and Na$_2$S (100 μM), **E** TT only (50 μM), **F** Na$_2$S$_2$ only (50 μM), **G** TT (50 μM) and Na$_2$S (100 μM), **H** Na$_2$S only (100 μM) or **I** no treatment for 30 min and washed twice with PBS. Cells were then treated with SSP4 (5 μM) and CTAB (100 μM) for 30 min and washed three times with PBS before being subjected to imaging. Scale bars, 100 μm. **J** Fluorescence intensities of SSP4 staining. Data are mean values ± SEM calculated using ImageJ. ($n$ = 9, where three different images were obtained from each of three distinct samples and quantified). Source data are provided as a Source Data file.

MS analysis (Supplementary Fig. 10), which indicated successful protein persulfidation. Together, these results confirm the efficiency of the TTS booster system in achieving protein persulfidation.

## Converting H$_2$S donors to H$_2$S$_2$ donors by TTS

So far, a large number of small molecule H$_2$S donors have been developed while known HSOH or H$_2$S$_2$ donors are still very limited. The unique bio-orthogonality of TTS suggested that TTS could be

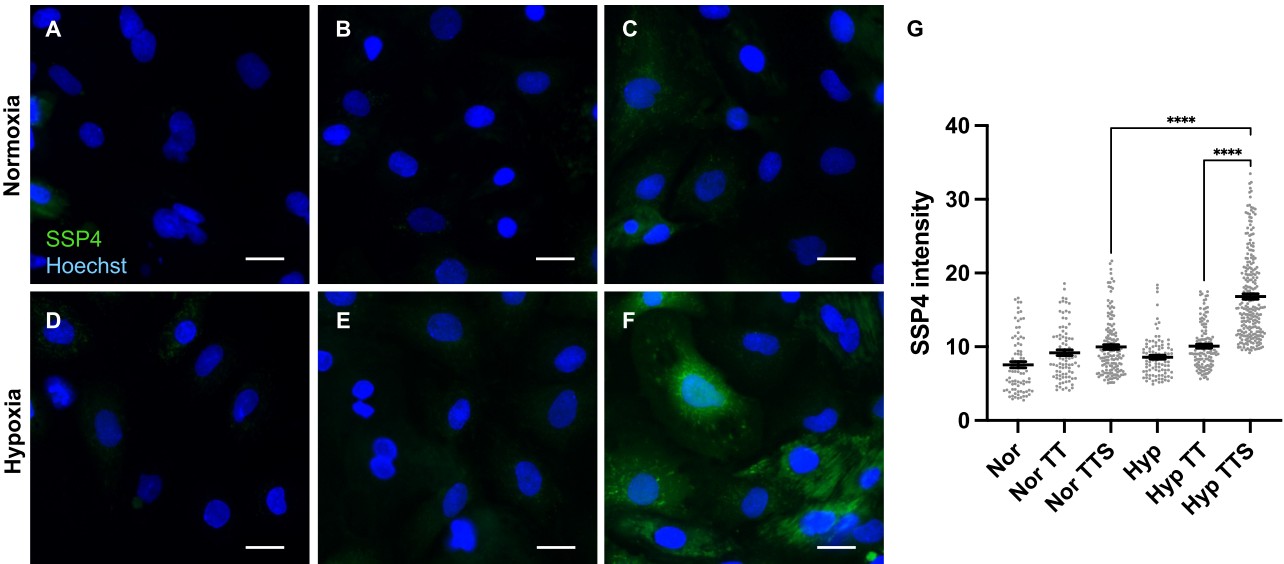

**Fig. 4 | Representative fluorescence images of normoxic and hypoxic neonatal rat cardiomyocytes.** Cells were incubated for 6 h under normoxia or hypoxia (1% $O_2$) and treated with (**A**, **D**) no treatment; (**B**, **E**) TT (10 μM); or (**C**, **F**) TTS (10 μM) during the last 30 min of normoxia or hypoxia. They were then incubated with SSP4 (5 μM) and Hoechst (2 μg/mL) for 30 min, washed three times with HBSS, and then imaged. Scale bars, 20 μm. **G** Data are mean values ± SEM calculated using ImageJ. (*n* = 84 (normoxia), 91 (normoxia TT), 152 (normoxia TTS), 96 (hypoxia), 134 (hypoxia TT) and 250 (hypoxia TTS) cells examined from 3 distinct samples). ****$P < 0.0001$; ordinary one-way ANOVA followed by Tukey's post-hoc test. Source data are provided as a Source Data file.

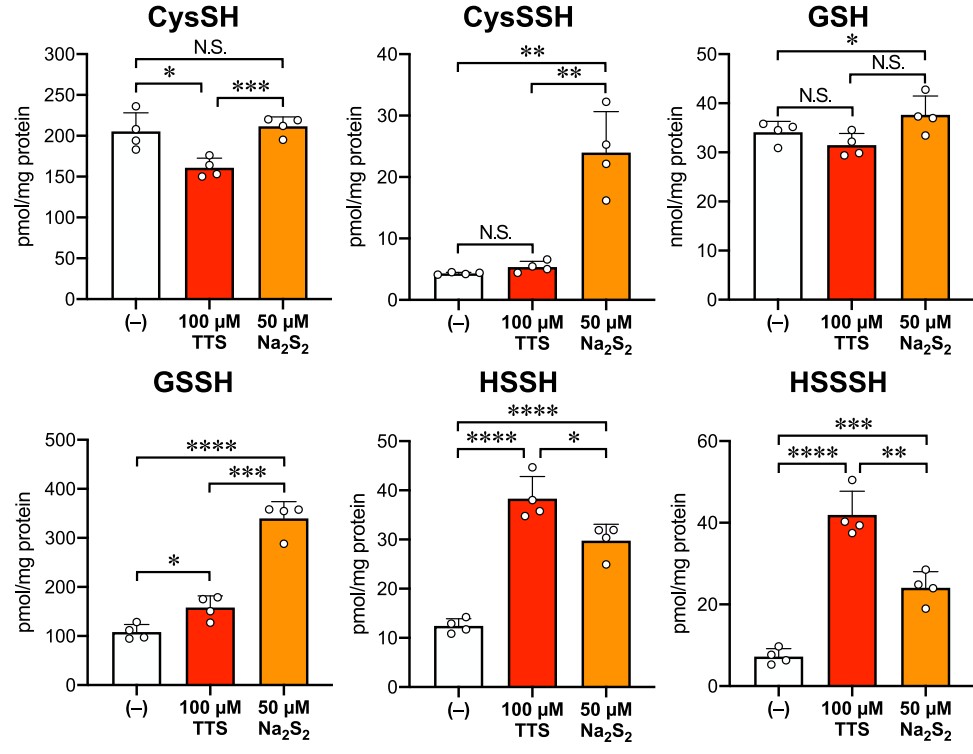

**Fig. 5 | The effect of TTS on the sulfur metabolome in MEFs.** WT MEFs were treated with 0, 100 μM TTS or 50 μM $Na_2S_2$ for 1 h at 37 °C, followed by sulfur metabolome analysis using HPE-IAM. Data are mean values ± S.D. (*n* = 4 samples). *$P = 0.014$, ***$P = 0.00085$ (CysSH), **$P = 0.0011$ (left in CysSSH), 0.0015 (right in CysSSH), *$P = 0.035$ (GSH), *$P = 0.013$, ***$P = 0.00013$, ****$P < 0.0001$ (GSSH), *$P = 0.022$, ****$P < 0.0001$ (HSSH), **$P = 0.0023$, ***$P = 0.00025$, ****$P < 0.0001$ (HSSSH). *$P < 0.05$, **$P < 0.01$, ***$P < 0.001$, ****$P < 0.0001$, N.S. (not significant); multiple *t*-test (two-sided). Source data are provided as a Source Data file.

used in conjunction with any $H_2S$ donor to create the corresponding HSOH/$H_2S_2$ donor system. We believe this will be a useful application of this compound. To prove this idea, we first tested the combination of TTS and JK-1, a hydrolysis-based $H_2S$ donor developed by our lab with demonstrated promising activities in cardiovascular disease models[42–44]. As shown in Fig. 7B, we envisioned that simply mixing JK-1 and TTS in solution would create HSOH/$H_2S_2$. To this end, the solution of JK-1 was prepared and then treated with TTS. After a 30 min incubation, the formation of HSOH/$H_2S_2$ was measured by SSP4. As shown in Fig. 7C, only the mixture of JK-1 and TTS

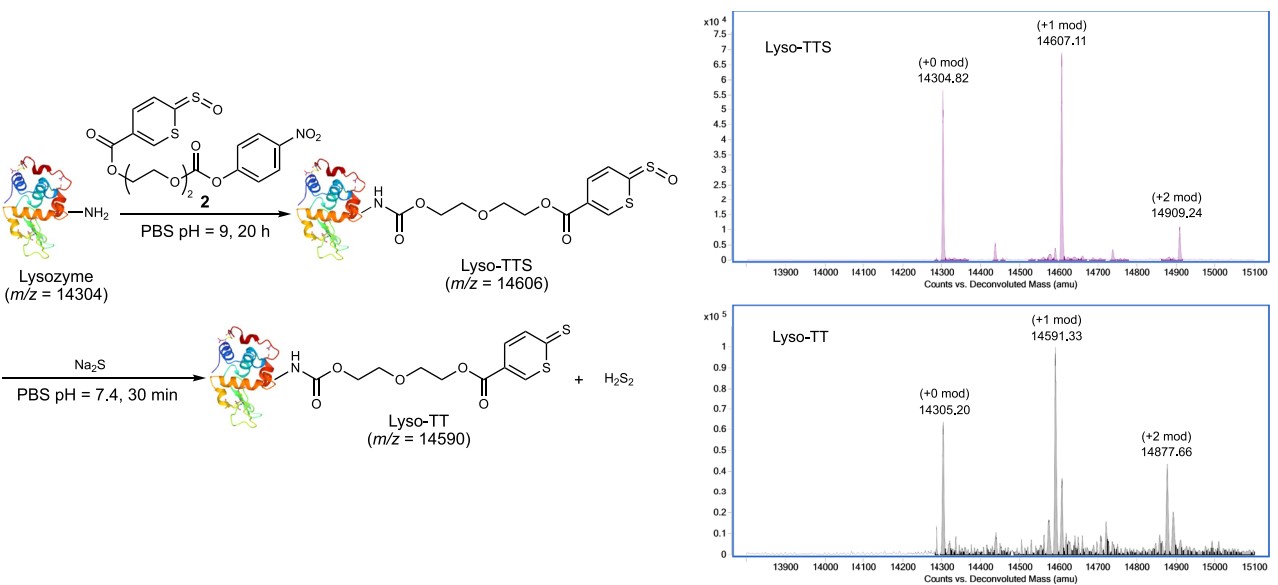

**Fig. 6 | Demonstration of TTS conjugation with proteins.** The reactions of TTS-lysozyme conjugate and the corresponding MS spectra.

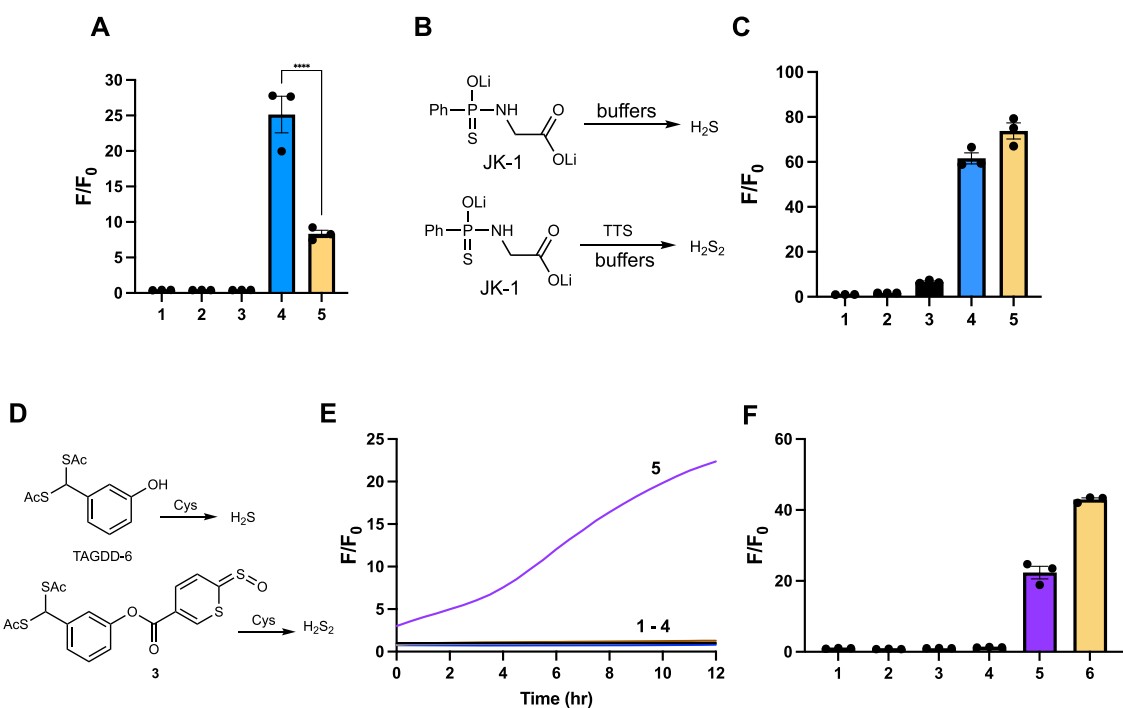

**Fig. 7 | TTS-mediated protein persulfidation and H$_2$S donor to H$_2$S$_2$ donor conversion. A** Fluorescence enhancement of SSP4 (5 μM) for the detection of persulfidated GAPDH (10 μM after Zeba column desalting and dilution) when the reduced protein (100 μM) was treated with: (1) control (no treatment); (2) Na$_2$S (1 mM); (3) TTS (1 mM); (4) TTS (1 mM) + Na$_2$S (1 mM); (5) Na$_2$S$_2$ (1 mM). Results are expressed as mean ± SEM (*n* = 3 distinct samples). Statistical analysis was performed using ordinary one-way ANOVA. ****$P < 0.0001$. **B** The reactions of JK-1 and JK-1 + TTS. **C** Fluorescence enhancements of SSP4 (40 μM) in the presence of: (1) SSP4 only; (2) TTS (200 μM); (3) JK-1 (400 μM); (4) TTS (200 μM) + JK-1 (400 μM); (5) Na$_2$S$_2$ (200 μM). Results are expressed as mean ± SEM (*n* = 3 distinct samples). **D** The reactions of TAGDD-6 and **3**. **E** Time-dependent fluorescence response of SSP4 over 12 h for 1) SSP4 only (5 μM); 2) **3** only (25 μM); 3) Cys only (100 μM); 4) TTS only (25 μM); 5) **3** (25 μM) + Cys (100 μM) and **F**) Fluorescence enhancement at the 12 h time point of 1) SSP4 only (5 μM); 2) **3** only (25 μM); 3) Cys only (100 μM); 4) TTS only (25 μM); 5) **3** (25 μM) + Cys (100 μM); 6) Na$_2$S$_2$ (25 μM). Compounds were first incubated in 50 mM PBS (pH 7.4) for 30 min in the dark at rt before SSP4 (5 μM) and CTAB (50 μM) were added and incubated for another 30 min. Results are expressed as mean ± SEM (*n* = 3 distinct samples). Source data are provided as a Source Data file.

gave strong fluorescence and was comparable to the signal obtained from Na$_2$S$_2$.

To further prove that TTS could turn an H$_2$S donor into an HSOH/H$_2$S$_2$ donor, we designed and synthesized a TTS-TAGDD hybrid compound **3** (Fig. 7D). TAGDD is a *gem*-dithiol based H$_2$S donor which can be triggered by biothiols such as cysteine to release H$_2$S[45]. We

expected that cysteine could still trigger the release of H$_2$S from the TAGDD component of **3** while the presence of TTS would allow for the formation of H$_2$S$_2$ (or its equivalent). As shown in Fig. 7E when **3** was treated with Cys, we observed steady and continuous H$_2$S$_2$ formation (monitored by SSP4 signals). Control experiments (e.g. **3** only or TTS only) did not give any notable fluorescence. Thus, the results shown in

Fig. 7 clearly demonstrated that TTS can boost $H_2S_2$ formation from $H_2S$ donors.

## TTS compensated for certain aspects of SQR deficiency

In cells, sulfide:quinone oxidoreductase (SQR) is one of the key enzymes responsible for $H_2S$ metabolism. SQR catalyzes the two-electron oxidation of $H_2S$ to form sulfane sulfurs ($S^0$) via persulfide and $H_2S_2$ intermediates[46–48]. Since TTS could promote the conversion of $H_2S$ to $H_2S_2$, we envisioned that TTS might be able to offset some features of SQR deficiency. To test this idea, cell imaging studies employing wild-type (WT) and SQR knockdown (SQR-KD) MEFs[49] were carried out. Briefly, the WT and SQR-KD MEFs were treated with several concentrations of $Na_2S$ or the mixture of $Na_2S$ and TTS in PBS for 20 min at 37 °C. Then, cellular sulfane sulfur levels were monitored and compared by fluorescence imaging with SSP4. As shown in Fig. 8, TTS was found to significantly increase intracellular sulfane sulfur levels, as evidenced by the elevation of fluorescence from both WT and SQR-KD MEFs treated with TTS (in an exogenously applied $Na_2S$ concentration-dependent manner). Intriguingly, such a $Na_2S$-enhanced fluorescence

was not observed with SQR-KD MEFs (Fig. 8A). The attenuated fluorescence signal by the SQR-KD was, however, almost completely restored by administering TTS to the SQR-KD cells in culture, as demonstrated by Fig. 8B. In addition, we also checked TTS's impact on mitochondrial function as assessed by the membrane potential formation through a JC-1 fluorescence imaging of the mitochondria. The results showed that TTS efficiently augmented the mitochondrial membrane potential of both WT and SQR-KD MEFs in a concentration-dependent manner (Fig. 8C). Notably, the membrane potential that was significantly impaired by SQR KD was found to be almost completely recovered by the TTS treatment to a level similar to that of the WT MEFs. The activity of TTS in this case might be explained by: 1) TTS could detoxify $H_2S$ by rapidly removing $H_2S$ from cellular media, and/or 2) the persulfide or $H_2S_2$ generated via the TTS reaction might mediate sulfur respiration, as proposed by several studies reported recently[11,48,50]. All these results suggest that TTS may be able to counteract some characteristics of SQR deficiency, and this might have some interesting applications. However, it should be noted that, unlike SQR, the reaction of TTS is not via a catalytic process. The regeneration

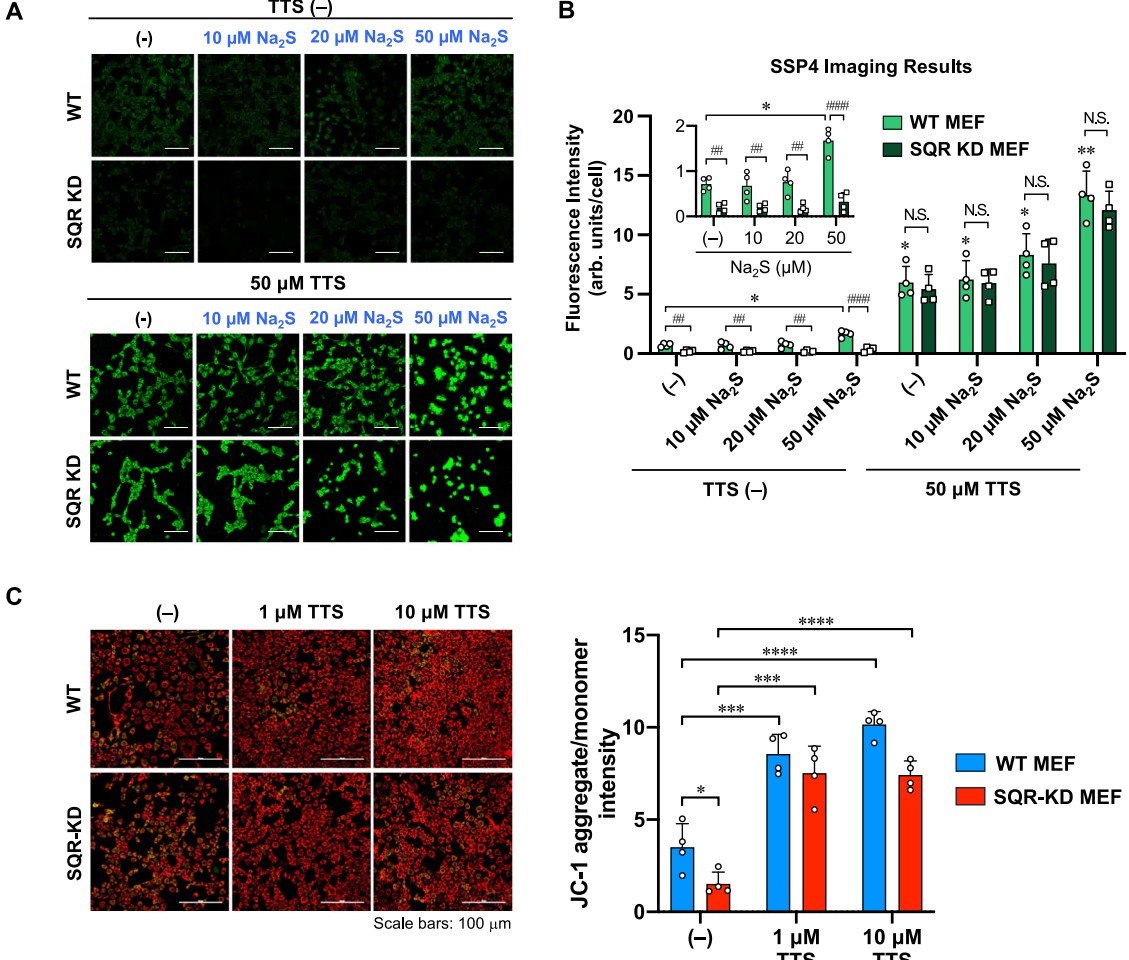

**Fig. 8 | The effects of TTS on sulfane sulfur generation and mitochondrial membrane potential in MEFs.** WT and SQR-KD MEFs were treated with 0, 10, 20, and 50 μM $Na_2S$ or a mixture of 50 μM TTS and 0, 10, 20, and 50 μM $Na_2S$ for 20 min at 37 °C and then subjected to SSP4 staining. **A** The fluorescence images and **B** fluorescence intensities of SSP4 staining. The inserted data (**B**) shows SSP4 fluorescence intensity in WT and SQR-KD cells treated with 0, 10, 20, and 50 μM $Na_2S$ without TTS. Scale bars, 100 μm. *$P = 0.0173, 0.0237, 0.0287, 0.0168$, **$P = 0.0038$ vs. TTS (-)/$Na_2S$ (-)/WT MEF; ##$P = 0.0047, 0.0099, 0.0019$, ####$P < 0.0001$; N.S., not significant. *$P < 0.05$, **$P < 0.01$; ##$P < 0.01$, ####$P < 0.0001$; two-way ANOVA with Tukey's test (two-sided). Data are mean

values ± S.D. from $n$ = four different images, each containing more than 50 cells in a representative experiment. Source data are provided as a Source Data file. **C** The effect of TTS on mitochondrial membrane potential in WT and SQR-KD MEFs. The cells underwent treatment with 0, 1, or 10 μM TTS for 30 min at 37 °C, followed by JC-1 staining. The fluorescence images (left panel) and fluorescence intensities (right panel) of JC-1 staining. Scale bars, 100 μm. *$P = 0.0312$, ***$P = 0.0009, 0.0003$, ****$P < 0.0001$. *$P < 0.05$, ***$P < 0.001$, ****$P < 0.0001$; multiple $t$-test (two-sided). Data are mean values ± S.D. from $n$ = four different images, each containing more than 100 cells in a representative experiment. Source data are provided as a Source Data file.

of TTS from TT by cellular oxidants like $H_2O_2$ is slow. Structural modifications on TTS may lead to compounds with faster regeneration kinetics, and this will be an interesting future research direction.

## Discussion

While the biological significance of $H_2S_2$ has been recognized in recent years, the study of $H_2S_2$ in biological systems is still a challenge due to the inherent instability of $H_2S_2$ and the lack of suitable donor compounds. In this work, we discovered *2H*-thiopyran-2-thione sulfine (TTS), a unique and redox regenerable booster compound that can efficiently convert $H_2S$ to $H_2S_2$. Mechanistic studies reveal that HSOH is the key intermediate from the reaction between TTS and $H_2S$. HSOH is also a highly reactive sulfane sulfur species, which is expected to behave like $H_2S_2$. So far, reliable methods for specific production of HSOH are still lacking. Thus, TTS may also be considered as a unique HSOH donor as well. The presence of biothiols would quickly react with HSOH to form persulfides, leading to the same result as that from $H_2S_2$. TTS could even be conjugated to proteins to achieve specific delivery for localized HSOH/$H_2S_2$ formation. We also demonstrated that the combination of TTS and $H_2S$ led to efficient protein persulfidation through the generation of $H_2S_2$ or its equivalents. Most interestingly, two examples were used to show that TTS could turn established $H_2S$ donors into $H_2S_2$ donors. In addition, TTS was found to be able to offset some effects of SQR deficiency in cells and promote the conversion of sulfide to sulfane sulfurs. These results demonstrate that TTS is a conceptually new strategy for the design of donor systems for $H_2S_2$ and/or its equivalents. Thus, we expect TTS to be a useful tool for elucidating the functions of $H_2S_2$, HSOH, and persulfides.

## Methods

### Syntheses and characterization of key substrates

TTS: To a stirred solution of methyl 2-thioxo-2*H*-thiopyran-5-carboxylate TT (186 mg, 1.0 mmol) in $CH_2Cl_2$ (10 mL), $NaHCO_3$ (84 mg, 5.0 mmol, 5.0 eq) and *m*-CPBA (70%, 246 mg, 1.0 mmol, 1.0 eq) were added. After being stirred at room temperature for 30 min, the reaction was quenched by sat. $NaHCO_3$ solution and extracted by ethyl acetate (EA). Combined organic layers were washed with brine and dried by $Na_2SO_4$. The solvent was removed under reduced pressure. The crude material was purified by flash column chromatography (HEX/EA = 5/1–3/1) to yield TTS (153 mg, yield = 76 %) as a dark red powder. TTS: $R_f$: 0.3 (HEX/EA = 2/1), m.p. = 130–131 °C. $^1$H NMR (600 MHz, $CD_2Cl_2$) δ 8.32 (dd, *J* = 1.2, 1.2 Hz, 1H), 7.06 (dd, *J* = 10.5, 1.2 Hz, 1H), 6.93 (dd, *J* = 10.5, 1.2 Hz, 1H), 3.85 (s, 3H). $^{13}$C NMR (151 MHz, $CD_2Cl_2$) δ 184.7, 163.3, 139.1, 126.8, 124.8, 120.2, 53.0. HRMS (ESI, *m/z*): $[M+H]^+$ calculated for $C_7H_7O_3S_2^+$: 202.9837; found: 202.9836.

TTS-Br: TTS-Br was prepared from S1 (see the SI) via esterification. $R_f$: 0.33 (HEX/EA = 2/1), m.p. = 163–165 °C. $^1$H NMR (600 MHz, DMSO-$d_6$) δ 9.00 (d, *J* = 1.3 Hz, 1H), 7.66 (d, *J* = 8.8 Hz, 2H), 7.52 (d, *J* = 10.5 Hz, 1H), 7.26 (d, *J* = 8.8 Hz, 2H), 7.02 (dd, *J* = 10.5, 1.3 Hz, 1H). $^{13}$C NMR (151 MHz, DMSO-$d_6$) δ 185.0, 160.9, 149.5, 142.2, 132.5, 124.9, 124.5, 124.2, 120.5, 118.6. HRMS (ESI, *m/z*): $[M]^+$ calculated for $C_{12}H_7BrO_3S_2^+$: 341.9020; found: 341.9050.

Compound **2**: To a stirred solution of S3 (see the SI) (165 mg, 0.39 mmol) in $CH_2Cl_2$ (5 mL), $NaHCO_3$ (164 mg, 1.95 mmol, 5.0 eq) and *m*-CPBA (70%, 123 mg, 0.5 mmol, 1.3 eq) were added. The mixture was stirred at room temperature for 10 min. Then, the reaction was quenched by sat. $NaHCO_3$ solution and extracted by EA. Combined organic layers were washed by brine and dried by $Na_2SO_4$. The solvent was removed by rotary evaporator. The crude material was purified by flash column chromatography (HEX/EA = 2/1–1/2) to yield **2** (105 mg, yield = 61 %) as a dark red powder. **2**: $R_f$: 0.5 (EA), m.p. = 81 – 83 °C. $^1$H NMR (600 MHz, $CD_2Cl_2$) δ 8.35 (s, 1H), 8.26 (d, *J* = 9.2 Hz, 2H), 7.39 (d, *J* = 9.2 Hz, 2H), 7.04 (d, *J* = 10.5 Hz, 1H), 6.93 (d, *J* = 10.5 Hz, 1H), 4.47–4.27 (m, 4H), 4.06–3.51 (m, 4H). $^{13}$C NMR (151 MHz, $CD_2Cl_2$) δ 184.6, 162.8, 155.9, 152.9, 145.9, 139.5, 126.7, 125.7, 124.7, 122.2, 120.2,

69.3, 69.0, 68.6, 65.0. HRMS (ESI, *m/z*): $[M+H]^+$ calculated for $C_{17}H_{16}NO_9S_2^+$: 442.0266; found: 442.0262.

Compound **3**: To a stirred solution of S5 (see the SI) (163 mg, 0.4 mmol) in $CH_2Cl_2$ (5 mL), $NaHCO_3$ (168 mg, 2.0 mmol, 2.0 eq) and *m*-CPBA (70%, 78 mg, 0.4 mmol, 1.0 eq) were added. After being stirred at room temperature for 10 min, the reaction was quenched by sat. $NaHCO_3$ solution and extracted by EA. Combined organic layers were washed by brine and dried by $Na_2SO_4$. The solvent was removed by rotary evaporator. The crude material was purified by flash column chromatography (HEX/EA = 3/1–2/1) to yield **3** (119 mg, yield = 70 %) as a dark red powder. **3**: $R_f$: 0.2 (HEX/EA = 1/1), m.p. = 97 – 98 °C. $^1$H NMR (600 MHz, $CD_2Cl_2$) δ 8.53 (dd, *J* = 1.2, 1.2 Hz, 1H), 7.41 – 7.34 (m, 2H), 7.28 – 7.23 (m, 1H), 7.12 (dd, *J* = 10.5, 1.2 Hz, 1H), 7.10 – 7.08 (m, 1H), 7.01 (dd, *J* = 10.5, 1.2 Hz, 1H), 6.20 (s, 1H), 2.32 (s, 6H). $^{13}$C NMR (151 MHz, $CD_2Cl_2$) δ 192.8, 184.2, 161.4, 150.8, 142.1, 140.8, 130.2, 126.12, 126.08, 124.5, 121.7, 121.3, 120.5, 47.7, 30.3. HRMS (ESI, *m/z*): $[M+H]^+$ calculated for $C_{17}H_{15}O_5S_4^+$: 426.9802; found: 426.9797.

### X-ray crystallography

Crystals of TTS-Br were grown from solvent diffusion. ~3 mg of solid TTS-Br was dissolved in ~2 mL of acetone and placed in a 1 dram glass vial. ~2 mL of pentane was layered on top and the vial was parafilmed. Dark brown-red needle-like crystals suitable for X-ray crystallography formed after approximately three weeks.

### SSP4 fluorescence assays

SSP4 was dissolved in DMSO to obtain a 200 μM stock solution. CTAB was dissolved in ethanol to yield a 2 mM stock solution. TTS and TT were dissolved in DMSO to make 1 mM stock solutions. $Na_2S$, $Na_2S_2$, L-Cys, and GSH were prepared in MilliQ $H_2O$ to obtain stock solutions at 2 mM, 1 mM, 4 mM or 8 mM (for the interference assay), and 40 mM (for the interference assay), respectively. 50 mM PBS buffer was added to individual wells of black polystyrene, flat-bottomed clear 96-well plate, followed by TTS (25 μM) or TT (25 μM), and $Na_2S$ (50 μM), cysteine (100 μM), or $Na_2S_2$ (25 μM). For the interference assay, TTS or TT (25 μM) were added to wells with or without $Na_2S$ (50 μM) next. Samples were mixed by pipetting up and down and incubated in the dark at room temperature (rt) for 30 min. Next, CTAB (50 μM) was added to all wells followed by SSP4 (5 μM), and samples were mixed by pipetting up and down. For the interference assay, Cys (200 μM) or GSH (1 mM) were first mixed with SSP4 (5 μM) and CTAB (50 μM) and then added to the appropriate wells. The final volume was 200 μL in each well. Samples were then incubated in the dark at rt for another 30 min before being measured on a Molecular Devices SpectraMax iD3 Multi-Mode Microplate Reader (emission 525 nm; excitation 485 nm; integration time 400 s, low PMT sensitivity, attenuation 1, read height 1.00 mm).

### Procedures for cell imaging

HeLa cells were obtained from ATCC (#CCL-2) and were seeded at ~7000 cells/well in a 96-well clear flat-bottomed BioLite plate (ThermoFisher Scientific, #130188) in DMEM/F12 (1:1) (Gibco, Invitrogen, #11330-032) medium supplemented with 10% fetal bovine serum (FBS) at 37 °C, 5% $CO_2$ overnight.

**Preparation before cell treatment.** $Na_2S$ and $Na_2S_2$ stocks were prepared at 10 mM in sterile 1X PBS. TTS and TT were prepared at 10 mM concentrations in DMSO. They were then diluted to 50 μM concentrations in serum-free Fluorobrite DMEM in individual Eppendorf tubes (0.5% DMSO). $Na_2S$ (final concentration = 100 μM) was added to the Eppendorf tube containing Fluorobrite DMEM and TTS or TT immediately before being added to the cells. Compounds were mixed by pipetting up and down prior to cell addition.

**Cell treatment.** Culturing media was aspirated from all wells, and cells were washed twice with 1X PBS. Compounds were added to cells (100 μL/well) and incubated for 30 min at 37 °C, 5% CO$_2$. While cells were incubating, SSP4 treatment was prepared by first adding 5 mM CTAB in ethanol to Fluorobrite followed by 6.125 mM SSP4 in DMSO to final concentrations of 100 μM and 5 μM, respectively, in a tube. The final volume was 5 mL (0.08% DMSO). Compounds were mixed by pipetting up and down prior to cell addition. After incubation, media was aspirated, and cells were washed twice with 1X PBS. SSP4 treatment was added to all wells (100 μL/well), and cells were incubated for 30 min at 37 °C, 5% CO$_2$ for 30 min. Media was then aspirated, and cells were washed three times with 1X PBS. Fluorobrite DMEM was added to all wells before imaging on the Keyence All-in-One Fluorescence Microscope (BZ-X810) (excitation: 470/40 nm; emission: 525/50 nm).

**Cell imaging under hypoxia and normoxia.** Neonatal rat cardiac myocytes (NRCMs) were prepared from the ventricles of 2-day-old SD rats[51]. All experiments using NRCMs were approved by the ethics committees at the National Institutes of Natural Sciences (Japan) and performed according to approved protocol code: 23A015. NRCMs were seeded on a matrigel-coated glass bottom dish and cultured in DMEM (low glucose) supplemented with 2% FBS. Two days after plating, NRCMs were incubated under normoxia or hypoxia (1% O$_2$) for 6 h. TT or TTS (10 μM) was added during the last 30 min of normoxia or hypoxia. Untreated cells served as controls.

For H$_2$S$_2$ imaging, NRCMs were incubated with 5 μM SSP4 in HBSS containing 0.04% Pluronic F-127 and 2 μg/mL Hoechst for 30 min[37]. NRCMs were washed with HBSS three times and imaged by the BZ-X710 (Keyence). The SSP4 fluorescence intensities from the cell images were calculated using Fiji and ImageJ (National Institutes of Health). Data are expressed as mean ± standard error of the mean (SEM). Statistical evaluations were performed on GraphPad Prism with ordinary one-way ANOVA.

For H$_2$S imaging, cells (without TT/TTS treatment) were incubated with 2.5 μM SF7-AM in HBSS containing 2 μg/mL Hoechst for 30 min. NRCMs were washed with HBSS three times and imaged by the Keyence BZ-X710. The results are shown in the Supplementary Fig. 8.

**Preparation of lysozyme-TTS conjugate**
To a 600 μL solution of chicken lysozyme (14.3 kDa, 300 μM in 1X PBS buffer, pH 9), **2** (30 μL, 60 mM in DMSO, 10 eq) was added. The mixture was incubated at room temperature for 16 h. Excess **2** was removed by a Zeba 7 K MWCO (ThermoFisher Scientific) spin column using PBS buffer pH 7.4 solution as the eluent. The final concentration of Lyso-TTS was 80 μM as determined by the Nanodrop One Spectrophotometer. Lyso-TTS was characterized by LC-MS (Fd. +0 mod: 14304.82; +1 mod: 14607.11; +2 mod: 14909.24).

**The reaction between lysozyme-TTS and H$_2$S**
To a solution of Lyso-TTS (80 μM, 300 μL in PBS buffer pH 7.4), Na$_2$S (30 μL, 10 mM in PBS pH 7.4 buffer) was added. The mixture was incubated at room temperature for 30 min to give Lyso-TT, which was then purified by a Zeba 7 K MWCO spin column using PBS buffer pH 7.4 as the eluent. The final concentration of the modified lysozyme (Lyso-TT) was 10 μM as determined by the Nanodrop One Spectrophotometer. Lyso-TT was characterized by LC-MS (Found. +0 mod: 14305.20; +1 mod: 14591.33; +2 mod: 14877.66).

**Computational method**
All calculations were performed with the Gaussian 09 program[52]. Geometry optimizations of all minima and transition states involved were carried out using M06-2X[53] functional and SMD[54] solvation model in H$_2$O solvent and the basis set was maug-cc-pVTZ[55]. It was labeled as SMD(H$_2$O)/M06-2X/maug-cc-pVTZ level. Frequency calculations at the same level were performed to validate each structure as either a minimum (the number of imaginary frequencies = 0) or a transition state (the number of imaginary frequencies = 1) and to evaluate its zero-point energy and thermal corrections at 298 K. Standard states are the hypothetical states at 1 mol/L.

**Fluorescence measurement of the reaction between JK-1 with TTS**
JK-1 (1.1 mg) was added to a 1 dram vial and 4.484 mL of 50 mM PBS (pH 6.0) was then added to obtain a 1 mM stock solution with no headspace in the vial. The solution was vortexed to mix and then incubated in the dark at rt for 20 min. Based on the H$_2$S-release profile reported[42] under the same conditions, 400 μM of H$_2$S (40% of 1 mM) should be released. The pH of the solution was then adjusted to ~7.4 by adding 16 μL 2 N NaOH. The same volume of 50 mM PBS (pH 6.0) buffer was added to an empty 1 dram vial and 16 μL of 2 N NaOH was added to adjust the pH of the buffer similarly for the controls. After incubation, 500 μL of the pH-adjusted PBS was aliquoted to nine 1.5 mL Eppendorf tubes while the same volume of JK-1 solution was aliquoted into six 1.5 mL Eppendorf tubes. TTS (10 mM stock in DMSO, 10 μL) was then added to three of the JK-1-containing tubes and to three of the pH-adjusted PBS only tubes. Na$_2$S$_2$ (10 mM stock in MilliQ H$_2$O, 10 μL) was added to three of the pH-adjusted PBS only tubes. Tubes were closed and vortexed to mix. Incubation occurred for 30 min at rt in the dark. CTAB (100 mM stock in EtOH, 2 μL) was then added to all 15 tubes followed by SSP4 (9.44 mM stock in DMSO, 2.11 μL). Tubes were closed, vortexed to mix, and incubated for 30 min in the dark at rt. A total of 200 μL of each tube was then added to separate wells on a 96-well black flat clear-bottomed plate before being measured on a Molecular Devices SpectraMax iD3 Multi-Mode Microplate Reader (emission 535 nm; excitation 485 nm; integration time 400 s, low PMT sensitivity, attenuation 1, read height 3.00 mm).

**Fluorescence measurement of the reaction of TTS-TAGDD hybrid compound 3**
TTS and **3** were prepared as 1 mM solutions in DMSO. L-cysteine and Na$_2$S$_2$ were prepared at 4 mM and 1 mM in MilliQ H$_2$O, respectively. SSP4 was dissolved in DMSO to obtain a 200 μM stock solution. CTAB was dissolved in ethanol to yield a 2 mM stock solution. 50 mM PBS buffer (pH 7.4) was added to individual wells of a black polystyrene, flat-bottomed clear 96-well plate, followed by TTS (25 μM) or **3** (25 μM) then cysteine (100 μM) or Na$_2$S$_2$ (25 μM). Samples were mixed by pipetting up and down and incubated in the dark at room temperature (rt) for 30 min. Next, CTAB (50 μM) was added to all wells followed by SSP4 (5 μM), and samples were mixed by pipetting up and down. Samples were then incubated in the dark at rt for another 30 min before being measured on a Molecular Devices SpectraMax iD3 Multi-Mode Microplate Reader (fluorescence mode; kinetics measurement type; emission 535 nm; excitation 485 nm; integration time 400 s, low PMT sensitivity, attenuation 1, read height 1.00 mm; total run time: 12 h; 25 cycles; interval: 0.5 h). Data was graphed as a time-dependent fluorescence response, with F$_0$ as the SSP4 only condition. The fluorescence of Na$_2$S$_2$ over time was not represented in the line graph presented due to the compound's known instability. Data obtained from the 12 h time point, including from Na$_2$S$_2$ was also represented in a bar graph for greater clarity. Results were expressed as mean ± SEM ($n$ = 3 distinct samples).

**Generation of SQR knockdown (KD) cell lines**
Generation of SQR-KD cell lines were according to the method we previously reported[49]. Briefly, short hairpin RNA sequences targeting mouse SQR were designed using BLOCK iT™ RNAi design software (Thermo Fisher Scientific). A pair of oligos for SQR shRNA, comprising sense: 5′-GATCCCCGGAGAGTTGGAGCAGAGAATGTTCAA-GAGACATTCTCTGCTCCAACTCTCCTTTTTA-3′ and anti-sense:

5′-AGCTTAAAAAGGAGAGTTGGAGCAGAGAATGTCTCTTGAA-CATTCTCTGCTCCAACTCTCCGGG-3′ were synthesized, annealed and ligated into the *Bgl* II and *Hind* III sites of pSUPER retro puro vector. MEFs were transfected with the pSUPER-shRNA (SQR) plasmid using Lipofectamine 3000. After 24 h, the medium was replaced, and cells were replated onto 10-cm dishes containing puromycin-supplemented medium. Puromycin-resistant clones were selected to establish stable SQR-KD cell lines.

## Imaging studies of TTS in the WT and SQR-KD MEFs

WT and SQR KD MEFs were seeded in eight-well glass chamber slides coated with poly-D-lysine (Sigma-Aldrich). The SQR-KD MEFs were produced according to the shRNA-mediated knockdown method reported recently[49]. The WT and SQR KD MEFs were treated with several concentrations of $Na_2S$ or mixture of $Na_2S$ and TTS in PBS for 20 min at 37 °C. The cells were washed once with PBS, followed by incubation with 20 μM SSP4 in PBS (+) containing 200 μM CTAB (cetyltrimethylammonium bromide) for 30 min at 37 °C. After removing the excess probes from the cells, cells were washed twice with PBS, followed by the measurement of fluorescence in images (excitation wavelength of 488 nm) with a confocal laser scanning microscope (Nikon C2 plus, NIS elements version 5.01 software). The fluorescence intensity of image was calculated as fluorescence intensity per cell (arb.units/cell) by using ImageJ software (National Institutes of Health) and dividing the intensity value by the number of cells.

## Measurements of mitochondrial membrane potential

To determine the mitochondrial membrane potential (ΔΨm) of MEFs under several experimental conditions, tetraethylbenzimidazolyl carbocyanine iodide (JC-1) staining was carried out following a previously reported method with some modifications[11]. The accumulation of the cell-permeable JC-1 probe (Abcam) within the mitochondria correlates with the membrane potential, which triggers a fluorescence emission shift from green to red. In brief, both WT and SQR-KD MEFs were cultured in 8-well slide chamber coated with poly-D-lysine and were treated with 1 or 10 μM TTS for 30 min at 37 °C. For JC-1 staining, the treated cells were washed with HKRB buffer (20 mM HEPES, 103 mM NaCl, 4.77 mM KCl, 0.5 mM $CaCl_2$, 1.2 mM $MgCl_2$, 1.2 mM $KH_2PO_4$, 25 mM $NaHCO_3$ and 15 mM glucose, pH 7.3), incubated with 20 μM JC-1 for 30 min at 37 °C, rinsed twice with HKRB buffer, and examined using a Nikon EZ-C2 Plus confocal laser microscope. ImageJ software was employed for image processing and quantification of the JC-1 fluorescence signals.

## Reporting summary

Further information on research design is available in the Nature Portfolio Reporting Summary linked to this article.

## Data availability

Data generated and analyzed in this study are included in this article and in its Supplementary Information files. The supplementary Cartesian Coordinates and MS files are included in the Supplementary Data 1 file. Source data are provided with this paper. The X-ray crystallographic coordinates for the structure reported in this work have been deposited at the Cambridge Crystallographic Data Center (CCDC). Accession code: CCDC 2259484. These data can be obtained free of charge via www.ccdc.cam.ac.uk/data_request/cif, or by emailing data_request@ccdc.cam.ac.uk, or by contacting The Cambridge Crystallographic Data Centre, 12 Union Road, Cambridge CB2 1EZ, UK; fax: +44 1223 336033. The protein mass spectrometry data have been deposited to the MassIVE repository with the dataset accession: MSV000094034. The data will also be available through the ProteomeXchange with the accession number: PXD049256. Source data are provided with this paper.

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

## Acknowledgements

This work was supported by the NIH (R01GM125968/R35GM149170 to M.X. and R01DK122160 to W.J.Q.), Grants-in-Aid for [Transformative Research Areas (A), International Leading Research, Scientific Research (S), (B) and (C) and Challenging Research (Exploratory)] from the Ministry of Education, Culture, Sports, Science and Technology (MEXT), Japan, to T.A. (21H05263, 22K19397, and 23K20040), T.M. (22K06893), J.M. (23K14341), S.O. (23K14333), M.M. (23K06145), and M.N. (21H05269, 22H02772, and 22K19395); Japan Science and Technology Agency (JST), CREST Grant Number JPMJCR2024 to M.N., T.A. and A.N., Japan; a grant from the Japan Agency for Medical Research and Development (AMED), Grant Number JP21zf0127001 to T.A. M.S. is supported by a NIH F31 Predoctoral Fellowship (F31HL170516). The NSF is gratefully acknowledged for its support of the acquisition of a single crystal X-ray diffractometer (CHE-2117549). DFT calculations were conducted using resources and services at the Center for Computation and Visualization (CCV) at Brown University. Part of the mass spectrometry experiments described herein were performed in the Environmental Molecular Sciences Laboratory, Pacific Northwest National Laboratory, a national scientific user facility sponsored by the Department of Energy under Contract DE-AC05-76RL0 1830.

## Author contributions

Q.C. and M.X. conceived the idea of the chemistry of TTS. Q.C., M.S., M.N., T.A. and M.X. designed the studies. Q.C. and T.W.P. carried out syntheses and mechanistic analyzes. Q.C. did DFT calculations. M.S. performed SSP4-based fluorescence assays on small molecules, proteins, HeLa cells, kinetic studies, and prepared the TTS-Br crystals. A.N. and M.N. performed NRCM-based studies. T.M. and T.A. performed MEF-based and mitochondrial membrane potential studies. S.S.K. and W.J.Q. performed LC-MS/MS studies. S.X. did kinetic studies. M.J. and M. M.

produced recombinant GAPDH and SQR-KD MEFs. S.O. and J.Y. performed sulfur metabolome MS analyzes. X.C. performed protein MS studies. J.R.R. performed X-ray crystallography analysis. All authors analyzed and interpreted data. Q.C., M.S., M.N., T.A. and M.X. prepared the figures and wrote and edited the paper. All authors reviewed the paper.

## Competing interests

The authors declare no competing interests.
