## [Peer Review File · Nature Communications]

2H-Thiopyran-2-Thione Sulfine, a Compound For Converting H₂S to HSOH/H₂S₂ and Increasing Intracellular Sulfane Sulfur LevelsREVIEWER COMMENTS

Reviewer #1 (Remarks to the Author):

It is increasingly evident that protein persulfidation plays important biological roles. H₂S₂ is known to easily react with thiols to yield persulfides. Thus, compounds that can generate H₂S₂ inside cells are of interest as chemical tools. Cui et al. present an elegant chemical mechanism and a compound (a sulfine) that converts H₂S into H₂S₂. This will be a potentially useful chemical tool to explore the biological effects of H₂S₂ generation and protein persulfidation.

It would seem important that the authors show that their compound (TTS) does indeed elevate H₂S₂ and persulfides inside living cells. They used SSP4 as an indicator for the intracellular formation of H₂S₂, but SSP4 is not a specific probe for H₂S₂. It would be more convincing to see MS-based relative quantitation (before and after TTS treatment) of alkylant-trapped H₂S₂, e.g., bismane-S-S-bismane. There is also no demonstration that protein persulfidation inside cells is increased by TTS treatment. Direct detection of intracellular protein persulfides would be useful. Additionally, H₂S₂ would also be expected to react with GSH to yield GSSH, which could also be trapped and quantitated by MS. One would expect GSSH to increase after TTS treatment.

How specific is TSS in reacting with H₂S? TTS did not show reactivity with Cys in vitro, but could it react with more nucleophilic (protein) thiols? The intermediary product is HSOH (hydrogen thioperoxide), which is supposed to react with another molecule of H₂S to yield H₂S₂. Isn't it possible for HSOH to engage in other reactions inside the cell? What is the reasoning that it exclusively reacts with H₂S? Could HSOH react with (protein) thiols to generate either a hydropersulfide or a sulfenic acid? Does cell treatment with TTS increase protein sulfenic acid levels, as for example detected by dimedone (or more advanced SOH reactive compounds)?

Figure 3, part II (HeLa cell experiment):

It would be helpful to also show untreated, Na₂S and TT+Na₂S. Is there an increase in the SSP4 signal in the TTS sample compared to untreated?

Fig. 3 part III:

Normoxic controls should be shown as well (same treatments). Also please define hypoxia conditions (%O₂?).

Its not clear to me what we can learn from the experiment using lysozyme-conjugated TTS. It is well established that H₂S₂, once formed, will persulfidate protein thiols in vitro. With a recombinant protein in vitro it should not matter if TTS is covalently attached to the protein or not. Persulfidation will happen anyway, once you generate H₂S₂. This in vitro experiment does not tell us if TTS can be used to perform location-specific protein persulfidation inside cells. This would need to be tested in a cellular context. How would you attach TTS to a specific protein inside a living cell?

That GAPDH thiols can be persulfidated by H₂S₂ (no matter how it is generated) would not be surprising. However, I am wondering if this experiment (Fig. 5 part I) really shows that TTS+Na₂S leads to GAPDH persulfidation. Is the desalting column really removing all the low molecular weight species? Even triple desalting may not be fully sufficient. Does the SSP4 signal really indicate persulfidated protein, or does it reflect residual H₂S₂ (or other S₀ species) that were not properly removed? To exclude the possibility of such artifacts, additional controls are needed: the same experiment without any protein, or using a protein without cysteines, or (ideally) a GAPDH mutant lacking cysteines.

Fig. 5, part III

Compound 3: Why is this compound not tried on cells? And then monitor the SSP4 signal over time? This would be interesting.

I think it is potentially misleading to say that TTS acts as a mimic of SQR. It seems to me that the only thing TTS and SQR have in common is that they can oxidize H₂S. However, SQR feeds electrons into the ETC and its product is not H₂S₂, as far as I understand it. Also, the experiment doesn't seem to show that TTS can replace SQR functionally. It's just increasing overall S₀ levels, so that smaller differences between WT and KO are not detectable any more. Seeing similar SSP4 signals does not mean that TTS functionally compensated for the lack of SQR.

Is there any evidence that TT is regenerated to TTS inside cells? Why is this called a “booster” system, because it can be regenerated in principle?

It is said in the introduction that 3MST and CARS produce H₂S₂. This sounds as if these enzymes directly generate H₂S₂ as a product. Is there any evidence for this? According to the literature, these enzymes primarily generate persulfides. H₂S₂ could just be a downstream product of persulfide speciation.

Reviewer #2 (Remarks to the Author):

This is a very interesting manuscript describing a unique approach to the generation of hydrogen persulfide (H₂S₂). Because there are very few viable approaches to H₂S₂ generation, this is a substantial advance. The approach involves the conversion of hydrogen sulfide (H₂S) to H₂S₂ via reaction with the sulfine TTS. Interestingly, the thione TT byproduct is shown to be re-oxidized to form TTS by a variety of biological oxidants. The system can be conjugated to proteins for localized H₂S₂ generation and also protein persulfidation is nicely demonstrated. On the whole, experiments are thorough and well executed. Publication with minor revisions is recommended with the following two questions for the authors to consider:

(1) H₂S₂ generation depends on the reaction of HSOH from TTS with H₂S. But what about reaction of HSOH with other nucleophiles? H₂S is not a tremendous nucleophile and is, for example, less nucleophilic than thiols. Wouldn't some (most?) trapping of HSOH by other biological nucleophiles be expected?

(2) In Figure 3, it seems that the observed intensity in IIB is much lower than that for the control Na₂S₂ in IID, presumably indicating quite low efficiency of H₂S₂ generation. Can these intensities be compared to provide an indication of the efficiency of H₂S₂ generation from the TTS + H₂S system?

Reviewer #3 (Remarks to the Author):

There are limited reports on H₂S₂ donors, and the manuscript by Cui et al. provides compelling results on a new strategy to generate H₂S₂ in cell cultures. The molecule synthesized in this work (TTS) is proposed to react with H₂S to release HSOH and a product (TT). If HSOH reacts with a second molecule of H₂S, H₂S₂ is formed. The authors affirm that endogenous oxidants can oxidize TT in cells to regenerate TTS. Therefore, if TTS is added to cell cultures along with an H₂S-donor, an H₂S₂ generating system could be established.

The authors explored the reaction mechanism in vitro and performed several experiments to generate H₂S₂ to persulfidate biological targets in cultures. The results obtained on this new approach are interesting and TTS could be a valuable tool for researchers keen on generating sulfane sulfur in solution. The text and the figures are clear, the methods section helps to understand how the experiments were performed, and the supporting information shows important controls. However, additional experiments could better support the authors' conclusions, particularly considering the complex environment that TTS would face in cell cultures.

Major concerns

1- As the authors highlight in the introduction, the presence of thiols could affect the efficiency of several donors and even change their identity. The role of thiols in the proposed H₂S₂-generating system is a question that remains unanswered in this manuscript.

- As far as I understood, TTS is a donor of HSOH. Once HSOH is released, it is able to react with several nucleophiles. I am particularly concerned about the thiols that are present in cells. While thiols are in millimolar levels, H₂S could hardly reach that concentration, even when H₂S-donors are present. The reaction of thiols with HSOH should produce persulfides. Then, TTS will still contribute to generating persulfides, not mediated by H₂S₂ but HSOH. This possibility should be mentioned to make readers aware of the chance of the direct persulfidation pathway without the intermediation of H₂S₂.

- The reaction of TTS with H₂S was nicely characterized using ³⁴S-labeled H₂S. Thiols, as well as H₂S, are also good nucleophiles. I wonder if the direct reaction of thiols with TTS could compete with the triggering reaction, maybe generating very different products. Is TTS stable in the presence of millimolar thiols? In Figure 3-I, the authors could include a condition where TTS, Na₂S, AND Cys (or other thiols) are incorporated to explore in vitro the

system's efficiency in the presence of thiols. The results shown in panels II and III are not completely illuminating since alternative pathways could generate sulfane sulfurs (e.g., H₂S oxidation by SQR, TTS degradation). Additionally, did the authors perform experiment controls in panel III under normoxic conditions? Was the H₂S₂ formation decreased?

2- The authors have proved the possibility of regenerating TTS from TT by oxidation.

However, the kinetics observed is not fast. Since the levels of the oxidants are not expected to be high in cells, recycling the donor seems slow and inefficient. Since the claim of the redox reversible booster system is highlighted even in the title of the manuscript, I am afraid that this conclusion should be better supported with experiments. In that way, did the authors try to persulfidate cell cultures by including TT (instead of TTS) with an H₂S donor? If no clear evidence of catalysis by TTS/TT is found, I strongly suggest moderating the conclusions regarding the system mimic of SQR.

Minor concerns

- Figure legend 1 A) Not only the novel H₂S₂ releasing system is shown here, mention the previous strategies.

- Line 100. Please help readers by providing a value for bond distance in interstitial SO₂.

- Line 106. How was this yield determined?

- Line 193. "The formation of H₂S₂ in this reaction was also confirmed." Please state how it was done.

- Lines 217-218. The advantage that the authors highlight here based on Figure 5I is unclear. Since no reference of 100% persulfidation is available, the conclusion may be that Na₂S₂ and the system TTS+Na₂S generate persulfides in GADPH at similar yields. Indeed, in Figure 5-II, a higher fluorescence emission is observed for Na₂S₂. This claim is resumed later in lines 285-286. By the way, in lines 536-539, the authors state that 5 μM SSP4 was incubated with 10 μM protein, could be the amount of SSP4 limiting?

- Figure legend 5-I. Please declare the concentration used.

- Please check references to figures along the text. Not every supplementary figure was cited.

- Page S6. The experimental conditions (eq.) indicated in the figure and the text are not the same. Please check. Check, also, the yield of TT-34S.

- Page S7. The 34S could be exchanged between labeled H₂S₂ molecules. I would expect the formation of H₂S₂ with two 34S, also. Did the authors find the product 1-34S with two

heavy sulfurs? Please provide a mass spectrum of the product 1-34S.

- Figure S4. The kinetic trace does not show a typical first-order behavior. How was the rate constant determined?

- Lines 245-247. Both HSOH and H₂S are able to spread to the bulk solution. The absence of other targets allows the formation of H₂S₂ more than the proximity of the chemical moieties.

- Lines 282-284. This issue was not assessed in this manuscript. Please provide references.

Response to Reviewers

We sincerely thank the reviewers for their very valuable comments. The following are the changes and our answers (which are colored in blue) to the reviewers:

Reviewer 1:

1. “It would seem important that the authors show that their compound (TTS) does indeed elevate H_2S_2 and persulfides inside living cells. They used SSP4 as an indicator for the intracellular formation of H_2S_2 , but SSP4 is not a specific probe for H_2S_2 . It would be more convincing to see MS-based relative quantitation (before and after TTS treatment) of alkylant-trapped H_2S_2 , e.g., bimane-S-S-bimane. There is also no demonstration that protein persulfidation inside cells is increased by TTS treatment. Direct detection of intracellular protein persulfides would be useful. Additionally, H_2S_2 would also be expected to react with GSH to yield GSSH, which could also be trapped and quantitated by MS. One would expect GSSH to increase after TTS treatment.”

Following this suggestion, we conducted sulfur metabolome analysis using HPE-IAM to quantitatively measure persulfides levels. The results (shown in Figure S9) revealed that TTS indeed increased persulfide levels such as GSSH, H_2S_2 and H_2S_3 , which is consistent with cell imaging results obtained with SSP4. These results have been added on page 10 in the manuscript.

2. “How specific is TSS in reacting with H_2S ? TTS did not show reactivity with Cys in vitro, but could it react with more nucleophilic (protein) thiols? The intermediary product is HSOH (hydrogen thioperoxide), which is supposed to react with another molecule of H_2S to yield H_2S_2 . Isn't it possible for HSOH to engage in other reactions inside the cell? What is the reasoning that it exclusively reacts with H_2S ? Could HSOH react with (protein) thiols to generate either a hydropersulfide or a sulfenic acid? Does cell treatment with TTS increase protein sulfenic acid levels, as for example detected by dimedone (or more advanced SOH reactive compounds)?”

We thank the reviewer for these important comments. To test if TTS could react with more nucleophilic (protein) thiols, we used reduced papain, which is a cysteine protease. Papain has one free cysteine in its active site (C25), with a pK_a value of ~ 3.3 . As shown in Figure S13 in the Supporting Materials, we did not observe significant changes to TTS. These results suggest that TTS was inert to protein thiols (this may not be definite, but at least for some proteins).

HSOH is a highly reactive species. We do expect it could react with cellular thiols (such as Cys or GSH). However, the products should be the corresponding hydropersulfides (RSSH), which should be the same products if H_2S_2 reacts with thiols. Therefore, the outcomes are the same. So, in reality, when TTS is treated with H_2S in the presence of thiols, elevated levels of persulfides (H_2S_2 or RSSH) should be produced and that is the ultimate goal of applying TTS. This was also confirmed by the results shown in Figure 3-I-B and the cell imaging studies.

As suggested by the reviewer, we thought about using dimedone to check if sulfenic acid levels could be increased by the TTS treatment. However, it is known that per/polysulfides can react with dimedone to form the dimedone products (likely via polysulfide hydrolysis) (Ref.: Sawa et al, *Antioxid. Redox Signal.*, 2021, doi.org/10.1089/ars.2021.0170). In our studies, we have observed that Na_2S_2 -treated proteins can lead to dimedone-adduct formation. Therefore, dimedone may not be a good reagent for our

studies. Nevertheless, this is an interesting question. We may consider working on this in our future work.

3. “Figure 3, part II (HeLa cell experiment): It would be helpful to also show untreated, Na₂S and TT+Na₂S. Is there an increase in the SSP4 signal in the TTS sample compared to untreated?”

We have added the images of these control studies (untreated, Na₂S only, TT+Na₂S). As shown in the revised Figure 3-II, these controls did not show obvious fluorescence. Only TTS+Na₂S gave significant fluorescence.

4. “Fig. 3 part III: Normoxic controls should be shown as well (same treatments). Also, please define hypoxia conditions (%O₂?).”

We thank the reviewer for this suggestion. We repeated cell imaging experiments under both hypoxia and normoxia. The updated results are shown in Figure 4. Clearly, hypoxia led to higher H₂S₂ production with TTS. We also updated the protocol in the Methods section. Our hypoxia condition was 1% O₂, and this is now noted in the caption of the figure.

5. “It is not clear to me what we can learn from the experiment using lysozyme-conjugated TTS. It is well established that H₂S₂, once formed, will persulfidate protein thiols in vitro. With a recombinant protein in vitro it should not matter if TTS is covalently attached to the protein or not. Persulfidation will happen anyway, once you generate H₂S₂. This in vitro experiment does not tell us if TTS can be used to perform location-specific protein persulfidation inside cells. This would need to be tested in a cellular context. How would you attach TTS to a specific protein inside a living cell?”

The reason we tested lysozyme-conjugated TTS was to demonstrate that TTS was sufficiently stable and could be used in relatively complex systems (such as under protein environments). There are methods to conjugate small molecule chemical probes to specific proteins. For example, using the genetically encoding method, one can install artificial groups (such as azides or alkynes) to specific proteins, which can then undergo ‘click chemistry’ to specifically link the small molecule probes. Based on the results from our lysozyme-conjugated TTS, we believe we can use similar strategies to conjugate TTS to a specific protein inside a living cell. Of course, such studies need time and collaborative efforts, and they are beyond the scope of this paper.

6. “That GAPDH thiols can be persulfidated by H₂S₂ (no matter how it is generated) would not be surprising. However, I am wondering if this experiment (Fig. 5 part I) really shows that TTS+Na₂S leads to GAPDH persulfidation. Is the desalting column really removing all the low molecular weight species? Even triple desalting may not be fully sufficient. Does the SSP4 signal really indicate persulfidated protein, or does it reflect residual H₂S₂ (or other S₀ species) that were not properly removed? To exclude the possibility of such artifacts, additional controls are needed: the same experiment without any protein, or using a protein without cysteines, or (ideally) a GAPDH mutant lacking cysteines.”

We thank the reviewer for raising the question about the efficiency of the desalting protocol in our experiments. To address this concern, we performed additional controls. As shown in Figure S10-A in the Supporting Information, we used -SH blocked GAPDH for our studies. This protein sample was treated with TTS+Na₂S or Na₂S₂ only, and then subjected to desalting by Zeba column (3 times). The solutions of TTS+Na₂S or Na₂S₂ without protein were also subjected to desalting by Zeba column. All these solutions were finally treated with SSP4 for fluorescence measurements. In all cases, we did not observe significant fluorescence (e.g. F/F₀ ≤ 1). To further confirm our results, we performed another experiment using recombinant human GAPDH proteins (both wild-type and with mutations at all three cysteine residues (C152/156/247S)). The proteins were prepared by site-directed mutagenesis. Both the wild-type and mutant GAPDH proteins underwent treatment without or with Na₂S, TTS, TTS+Na₂S, or Na₂S₂, followed by desalting using the G-25 column and then fluorescence measurements by the SSP4

assay. The results (shown in Figure S10-B) demonstrate that, while wild-type GAPDH significantly increased SSP4 fluorescence with TTS+Na₂S or Na₂S₂ treatments, the cysteine mutant showed no increase in SSP4 fluorescence. All these results make us to believe that the desalting step was efficient, and the observed signal (in Figure 6-I) was due to persulfidation.

7. “Fig. 5, part III: Compound 3: Why is this compound not tried on cells? And then monitor the SSP4 signal over time? This would be interesting.”

We thank the reviewer for this suggestion. In fact, we have attempted to test compound **3** in cells but the results were not informative. As shown in Figure 6-III, the production of sulfane sulfurs from compound **3** in the presence of cysteine is a slow process (requires many hours). We found that cells treated with **3** for that long did not look healthy. While we did observe increased fluorescence with SSP4, we could not perform control studies as the depletion of free thiols in cells for such a long period of time would completely damage the cells.

8. “I think it is potentially misleading to say that TTS acts as a mimic of SQR. It seems to me that the only thing TTS and SQR have in common is that they can oxidize H₂S. However, SQR feeds electrons into the ETC and its product is not H₂S₂, as far as I understand it. Also, the experiment doesn’t seem to show that TTS can replace SQR functionally. It’s just increasing overall S0 levels, so that smaller differences between WT and KO are not detectable any more. Seeing similar SSP4 signals does not mean that TTS functionally compensated for the lack of SQR.”

To address this comment, we performed new experiments to check TTS’s impact on mitochondrial function as assessed by the membrane potential formation through a JC-1 fluorescence imaging of the mitochondria. The results showed that TTS efficiently augmented the mitochondrial membrane potential of both WT and SQR-KD MEFs in a concentration-dependent manner (Figure 7-C). These results suggest that TTS could somehow exert SQR-like activities, and this might have some interesting applications. However, we agree with the reviewer that it may not be appropriate to call TTS a mimic of SQR as the reaction of TTS is not through a catalytic process. The regeneration of TTS from TT by cellular oxidants like H₂O₂ is slow. Nevertheless, structural modifications on TTS may lead to compounds with faster regeneration kinetics, and this will be an interesting topic for our follow-up studies. To avoid any confusion, we have removed ‘SQR-mimic’ from the manuscript. Instead, we just said ‘TTS has some SQR-like activities.’

9. “Is there any evidence that TT is regenerated to TTS inside cells? Why is this called a “booster” system, because it can be regenerated in principle?”

As shown in our kinetic studies, the oxidation of TT to form TTS by biologically relevant oxidants, especially H₂O₂, is a slow reaction. We do not yet have evidence to show TT could reform TTS inside cells but that is perhaps possible (at least theoretically). We call it ‘booster’ simply because TTS could covert H₂S to H₂S₂, thereby boosting its reactivity. If the reviewer feels the name ‘booster’ is misleading, we are willing to remove it from the manuscript.

10. “It is said in the introduction that 3MST and CARS produce H₂S₂. This sounds as if these enzymes directly generate H₂S₂ as a product. Is there any evidence for this? According to the literature, these enzymes primarily generate persulfides. H₂S₂ could just be a downstream product of persulfide speciation.”

The reviewer is correct. These enzymes do not directly generate H₂S₂ as the product. To avoid any confusion, we revised the sentence to: “*Endogenous H₂S₂ is produced indirectly by enzymes like 3-mercaptopyruvate transferase (3-MST) and cysteinyl-tRNA synthetase (CARS), via persulfides as the key intermediates.*”

Reviewer 2:

1. "H₂S₂ generation depends on the reaction of HSOH from TTS with H₂S. But what about reaction of HSOH with other nucleophiles? H₂S is not a tremendous nucleophile and is, for example, less nucleophilic than thiols. Wouldn't some (most?) trapping of HSOH by other biological nucleophiles be expected?"

HSOH is a highly reactive species. We do expect that it could react with cellular thiols (such as Cys or GSH). However, the products should be the corresponding hydropersulfides (RSSH), which should be the same products if H₂S₂ reacts with thiols. Therefore, the outcomes are the same. So, in reality, when TTS is treated with H₂S in the presence of thiols, elevated levels of persulfides (H₂S₂ or RSSH) should be produced and that is the ultimate goal of applying TTS. This has been confirmed in our studies.

2. " In Figure 3, it seems that the observed intensity in IIB is much lower than that for the control Na₂S₂ in IID, presumably indicating quite low efficiency of H₂S₂ generation. Can these intensities be compared to provide an indication of the efficiency of H₂S₂ generation from the TTS + H₂S system?"

We repeated the cell imaging experiments shown in Figure 3-II. We added three additional control experiments suggested by Reviewer #1. We also provided a comparison of fluorescence intensities in Figure 3 II-H using ImageJ. While the intensity of the Na₂S₂ treatment was still higher than that of TTS+H₂S, we feel the signal from TTS+H₂S was significant, indicating a high efficiency of H₂S₂ generation from this system.

Reviewer 3:

1. "As the authors highlight in the introduction, the presence of thiols could affect the efficiency of several donors and even change their identity. The role of thiols in the proposed H₂S₂-generating system is a question that remains unanswered in this manuscript. As far as I understood, TTS is a donor of HSOH. Once HSOH is released, it is able to react with several nucleophiles. I am particularly concerned about the thiols that are present in cells. While thiols are in millimolar levels, H₂S could hardly reach that concentration, even when H₂S-donors are present. The reaction of thiols with HSOH should produce persulfides. Then, TTS will still contribute to generating persulfides, not mediated by H₂S₂ but HSOH. This possibility should be mentioned to make readers aware of the chance of the direct persulfidation pathway without the intermediation of H₂S₂."

We agree with the reviewer that TTS can be considered as a donor of HSOH. Once HSOH is formed, it can react with thiols to produce persulfides. This leads to the same results as when H₂S₂ is generated from H₂S and biothiols are also present. In both cases, persulfides are the major products (as demonstrated in our cell studies). As suggested by the reviewer, we added the following statement in the *Discussion* to clarify this point. "Our mechanistic studies reveal that HSOH is the key intermediate from the reaction between TTS and H₂S. Thus, TTS may also be considered as a HSOH donor. The presence of biothiols would quickly react with HSOH to form persulfides, leading to the same result as that from H₂S₂. This unique feature should make TTS a useful tool for elucidating the functions of H₂S₂ and persulfides, and promoting related therapeutic applications."

2. "The reaction of TTS with H₂S was nicely characterized using ³⁴S-labeled H₂S. Thiols, as well as H₂S, are also good nucleophiles. I wonder if the direct reaction of thiols with TTS could compete with the triggering reaction, maybe generating very different products. Is TTS stable in the presence of millimolar thiols? In Figure 3-I, the authors could include a condition where TTS, Na₂S, AND Cys (or other thiols) are incorporated to explore in vitro the system's efficiency in the presence of thiols. The results shown in panels II and III are not completely illuminating since alternative pathways could generate sulfane sulfurs (e.g., H₂S oxidation by SQR, TTS degradation)."

We thank the reviewer for bringing up this important question about the possible interference of thiols on the triggering reaction between TTS and H₂S. As many papers have demonstrated, thiols (RSH) can

react with sulfane sulfurs (H_2S_n , $\text{R}'\text{SSH}$, or $\text{R}'\text{SOH/HSOH}$) to form disulfides, persulfides, etc. Eventually, complex dynamic equilibriums with different sulfur species (disulfides, H_2S , RSSH , etc) would be formed no matter how the initial sulfane sulfur species is generated. We feel that what matters is the novel method to promote sulfane sulfur formation by TTS. The results from our cell-based studies, cysteine-triggered H_2S_2 donors (Fig. 6-III), and the corresponding control studies all confirmed that TTS could effectively induce sulfane sulfur production. Based on the suggestion from this reviewer, we have added interference studies to Figure 3-I. As shown in the revised Figure 3-I-B, the presence of thiols was found to cause somewhat decreased fluorescence. This was expected as thiols could compete with SSP4 to react with H_2S_2 , therefore causing decreased signals. Nevertheless, the observed fluorescence was still significant. In addition, our cell imaging studies (Figure 3-II, Figure 4) and HPE-IAM trapping studies (Figure S9) provide further evidence to show that TTS could effectively react with H_2S to form H_2S_2 (or persulfides) in the presence of thiols (as thiols are ubiquitously present in cells).

3. “Additionally, did the authors perform experiment controls in panel III under normoxic conditions? Was the H_2S_2 formation decreased?”

We repeated the cell imaging experiments under both hypoxia and normoxia. The updated results are now shown in Figure 4. Clearly, hypoxia led to higher H_2S_2 production with TTS. We have also updated the protocol in the Methods section. Our hypoxia condition was 1% O_2 , and this information has now been added to the figure caption.

4. “The authors have proved the possibility of regenerating TTS from TT by oxidation. However, the kinetics observed is not fast. Since the levels of the oxidants are not expected to be high in cells, recycling the donor seems slow and inefficient. Since the claim of the redox reversible booster system is highlighted even in the title of the manuscript, I am afraid that this conclusion should be better supported with experiments. In that way, did the authors try to persulfidate cell cultures by including TT (instead of TTS) with an H_2S donor? If no clear evidence of catalysis by TTS/TT is found, I strongly suggest moderating the conclusions regarding the system mimic of SQR.”

The reviewer is correct. The oxidation of TT to TTS by biologically relevant oxidants (such as H_2O_2) is slow and may not be feasible under normal conditions. We do not yet have evidence to show that TT could reform TTS inside cells. The regeneration is just theoretically possible. Based on this comment, we moderated the conclusions regarding the system as a mimic of SQR. We can even change the title to “Converting H_2S to H_2S_2 by a Redox Regenerable Heterocyclic Sulfine Compound” if the reviewer feels that is appropriate.

Minor Concerns:

5. “Figure legend 1 A) Not only the novel H_2S_2 releasing system is shown here, mention the previous strategies.”

We have added ‘Reported H_2S_2 donors’ to Figure legend 1-A.

6. “Line 100. Please help readers by providing a value for bond distance in interstitial SO_2 .”
The bond distance value ($\sim 1.380 \text{ \AA}$) has been provided.

7. Line 106. How was this yield determined?

This yield (80%) is the isolated yield from the reaction. We have indicated it in the text.

8. “Line 193. “The formation of H_2S_2 in this reaction was also confirmed.” Please state how it was done.”
The formation of H_2S_2 was confirmed by the SSP4 assay. This information has been added to the text.

9. Lines 217-218. The advantage that the authors highlight here based on Figure 5I is unclear. Since no reference of 100% persulfidation is available, the conclusion may be that Na₂S₂ and the system TTS+Na₂S generate persulfides in GAPDH at similar yields. Indeed, in Figure 5-II, a higher fluorescence emission is observed for Na₂S₂. This claim is resumed later in lines 285-286. By the way, in lines 536-539, the authors state that 5 μM SSP4 was incubated with 10 μM protein, could be the amount of SSP4 limiting?

Based on the comment from Reviewer #1, we repeated the GAPDH persulfidation experiments and compared the effects of TTS+Na₂S vs Na₂S₂. We made sure to remove small molecule reagents from the protein samples by desalting three times. As shown in Figure 6-I, the fluorescent signals obtained from TTS+Na₂S were significantly stronger than those obtained from Na₂S₂. These results indicate that TTS+Na₂S was very effective in protein persulfidation. We agree with this reviewer that in some situations (for example, in cell imaging studies), Na₂S₂ might be more effective than TTS+Na₂S in persulfidation. To avoid any confusion, we decided not to compare TTS+Na₂S with Na₂S₂. We revised the sentences (as indicated by the reviewer) and simply indicated '*TTS+Na₂S is an effective way for persulfidation.*' We used 5 μM SSP4 for protein persulfidation measurements as this concentration was found to be optimal for protein studies in our previous studies (ref. 34).

10. Figure legend 5-I. Please declare the concentration used.

The concentrations of the chemicals are now provided in the Figure legend (now Figure 6-I).

11. Please check references to figures along the text. Not every supplementary figure was cited.

We have checked this problem and made sure to cite all supplementary figures that needed to be cited in the manuscript. Some supplementary figures are only included to support the data shown in the Supporting Information, so they are cited in the SI but not in the main text.

12. Page S6. The experimental conditions (eq.) indicated in the figure and the text are not the same. Please check. Check, also, the yield of TT-³⁴S.

We have corrected the typo in the text. It is 4.0 equivalents. We mistakenly put in the wrong volume '2 mL'. It should be '6.7 mL'. The yield of TT-³⁴S was confirmed to be 52%.

13. Page S7. The ³⁴S could be exchanged between labeled H₂S₂ molecules. I would expect the formation of H₂S₂ with two ³⁴S, also. Did the authors find the product 1-³⁴S with two heavy sulfurs? Please provide a mass spectrum of the product 1-³⁴S.

We agree with the reviewer that the formation of H₂S₂ with two ³⁴S atoms is theoretically possible. However, we did not see the obvious mass peak corresponding to the trapped product with two heavy sulfurs. Perhaps under our conditions, the trapping reaction is much faster than the sulfur exchange reaction. The mass spectrum of 1-³⁴S has been added to the Supporting Information (Figure S5).

14. Figure S4. The kinetic trace does not show a typical first-order behavior. How was the rate constant determined?

The original Figure S4 showed UV-Vis absorption changes at 380 nm and 339 nm. Since both TTS and TT exhibit absorption at these two wavelengths, their overlap signals likely lead to non-typical first order traces. To avoid any confusion, we have replaced that figure with a new figure (Figure S7) showing time-dependent concentration changes of TTS. At each time point, the concentration of TTS was calculated based on the equation shown on page S13. The kinetic trace follows a typical first-order behavior. We have repeated all kinetic experiments and confirmed the data.

15. Lines 245-247. Both HSOH and H₂S are able to spread to the bulk solution. The absence of other targets allows the formation of H₂S₂ more than the proximity of the chemical moieties.

We thank the reviewer for pointing this out. Based on this comment, we have removed 'close proximity' from the sentence.

16. Lines 282-284. This issue was not assessed in this manuscript. Please provide references. We agree with the reviewer that it is premature to claim the bio-orthogonality of this reaction. Clearly more studies are needed to clarify this. So, we have decided to remove this sentence from the manuscript.

REVIEWER COMMENTS

Reviewer #1 (Remarks to the Author):

The authors have done a very good job in providing additional experiments/controls and clarifications. The experimental part of the manuscript has been improved substantially.

However, I am still somewhat uncomfortable with how the authors present their results in writing. I consider it important to make a clear distinction between what is actually shown and what is interpretation or even speculation. I feel that the authors should be more critical about their work, more careful in their wording, and should openly discuss the limitations and unknowns of their study.

The most important question is what TTS is doing inside cells. The paper provides solid evidence that TTS increases levels of intracellular sulfane sulfur (S₀) species. A key piece of evidence for this is the sulfur metabolite analysis (why is this “hidden” in the supplement?). Since there is no analysis of protein persulfidation in cells before and after TTS treatment, the paper cannot make strong claims about this point.

Strictly speaking, we do not know if the proposed mechanism really applies to the intracellular situation. I would agree that it is practically certain that HSOH is generated inside cells, in an H₂S dependent manner. But we cannot be certain about any of the following steps. Perhaps (almost) all the HSOH reacts with thiols to persulfides. Perhaps the H₂S₂ detected by metabolomics is a downstream secondary product of further speciation. If true, this may not make a practical difference for those who want to use the compound, but it makes a substantial difference for how to present and discuss the results. It doesn't seem appropriate to just mention this possibility as the last sentence of a (very short) discussion (that otherwise lacks any critical discussion). Along these lines, why would HSOH not react with thiols to generate sulfenic acids? I understand this issue is difficult to address experimentally. What is the chemical argument here? This should be part of the discussion section. The current discussion is insufficient, it should discuss in some detail the potential problems and unknowns of the approach.

The abstract is inaccurate, if not misleading, in my opinion. The system is not as simple as presented. The abstract should expose to the reader that TTS primarily creates HSOH and that there are several possibilities of how it secondarily creates S0 species in cells (including H2S2). TTS is an HSOH donor (not an H2S2 donor!), and this should be clearly communicated from the start. Selling the compound as a H2S to H2S2 converter for cells is not fully appropriate in my opinion. It is also misleading to claim that localized H2S2 formation was achieved when in fact this experiment was only conducted in vitro with a recombinant protein (where localized production doesn't play a role). In my opinion, ascribing SQR-like activity to TTS is at least confusing, if not misleading. The activity/mechanism of SQR is very different from TTS. We can only say that TTS can compensate for certain aspects of SQR deficiency.

The title should be changed, because it is misleading, in my opinion. There is no demonstration of a "redox-reversible booster system" in this paper. This is a speculation that can be and should be part of the discussion section, but not of the title. In my opinion, a more appropriate title would be something like this: "A compound converting H2S to HSOH increases intracellular sulfane sulfur levels".

Reviewer #2 (Remarks to the Author):

I believe that the comments/concerns from the previous reviews have been adequately addressed by the authors and the manuscript is now suitable for publication.

Reviewer #3 (Remarks to the Author):

The authors performed essential controls that strengthened the main conclusions of the manuscript and clarified my concerns. This work will be an exciting contribution to the community working on sulfur biochemistry. However, there are a few details that the authors may consider checking in the final version of the manuscript:

- HRMS spectra in Figure S5 show additional peaks that seem interesting. Close to the peak of the product TT (m/z 186.9888), a peak with slightly higher intensity is detected (m/z 188.98550), which is compatible with TT with one heavy sulfur. Additionally, the peak of 1-

34S (215.96038) is close to one (213.96278), compatible with compound 1 without heavy sulfurs. While the latter is expected to be formed by sulfur exchange among H₂S₂ molecules in solution, the peak of TT molecules that conserves the heavy atom is not easily predicted according to the mechanism proposed. Is it possible that this peak appears due to natural isotope distribution? Do the authors consider that alternative reactions could be happening? Could the yield shown on page S9 (52%) be related to this observation?

· Page 10. Lines 200-207. Here, the authors present results on H₂S₂ detection by MS after treatment of MEFs with TTS alone. To help the readers, the authors may mention that later - in Figure 7- it is shown that TTS alone can increase sulfane sulfur in MEFs. While following the thread of the text, up to this paragraph, only combinations of TTS and H₂S donors or treatment with TTS under hypoxic conditions –and not TTS alone- were able to increase sulfane levels significantly (HeLa cells in Figure 3 and NRCMs in Figure 4).

· Figure 6-III-A. A minor observation: trace "6" is lacking in the plot. I wonder if fluorescence in trace "5" shows a slow increase due to a slow H₂S₂ formation or a slow reaction of H₂S₂ with SSP4.

· Figure 7-C. The new results on mitochondrial membrane potential that the authors provided are interesting and lead to fundamental questions that groups working in mitochondria biochemistry would find appealing. However, as discussed by the authors, possible causes of TTS's effect on mitochondrial membrane potential are not yet clear. Therefore, the mention of the similarity of TTS to SQR weakens a solid work since the data do not sufficiently support it: 1- the theoretical products of the reaction of TTS with H₂S - HSOH or H₂S₂- are not produced by SQR (only downstream indirect products could be similar), 2- according to the kinetic data and the culture treatment with TT, the activity of TTS is not catalytic like SQR's, and 3- the cause of the effect of TTS on mitochondrial membrane potential is not yet clear.

· Lines 298-301. It is not clear to this reviewer why persulfides would accumulate in the mitochondria. TTS could consume H₂S in the cytosolic compartment, too. Maybe the overall decrease in H₂S is simply responsible for the effect observed.

Response to Reviewers

We thank the reviewers for reviewing our revised manuscript. We are glad that the reviewers found our revision acceptable. Based on their additional comments, we have further revised the manuscript. The following are the changes and our answers (which are colored in blue) to the reviewers:

Reviewer 1:

1. “The authors have done a very good job in providing additional experiments/controls and clarifications. The experimental part of the manuscript has been improved substantially.”

We thank the reviewer for this positive comment.

2. “However, I am still somewhat uncomfortable with how the authors present their results in writing. I consider it important to make a clear distinction between what is actually shown and what is interpretation or even speculation. I feel that the authors should be more critical about their work, more careful in their wording, and should openly discuss the limitations and unknowns of their study.”

We appreciate these comments. We have further revised the manuscript to make sure our discussions/words are appropriate. Please see those highlighted in yellow.

3. “The most important question is what TTS is doing inside cells. The paper provides solid evidence that TTS increases levels of intracellular sulfane sulfur (S₀) species. A key piece of evidence for this is the sulfur metabolite analysis (why is this “hidden” in the supplement?). Since there is no analysis of protein persulfidation in cells before and after TTS treatment, the paper cannot make strong claims about this point.”

We have moved the figure showing the sulfur metabolite results to the main text of the manuscript (now Figure 5). We have revised the subtitle of this section to “Quantitative determination of intracellular sulfane sulfur levels with TTS treatment”.

4. “Strictly speaking, we do not know if the proposed mechanism really applies to the intracellular situation. I would agree that it is practically certain that HSOH is generated inside cells, in an H₂S dependent manner. But we cannot be certain about any of the following steps. Perhaps (almost) all the HSOH reacts with thiols to persulfides. Perhaps the H₂S₂ detected by metabolomics is a downstream secondary product of further speciation. If true, this may not make a practical difference for those who want to use the compound, but it makes a substantial difference for how to present and discuss the results. It doesn’t seem appropriate to just mention this possibility as the last sentence of a (very short) discussion (that otherwise lacks any critical discussion). Along these lines, why would HSOH not react with thiols to generate sulfenic acids? I understand this issue is difficult to address experimentally. What is the chemical argument here? This should be part of the discussion section. The current discussion is insufficient, it should discuss in some detail the potential problems and unknowns of the approach.”

We understand the reviewer’s concern here and have accordingly revised the manuscript to clarify the point. We have added more discussions throughout the manuscript to discuss the importance of HSOH and how TTS is linked to HSOH and H₂S₂. Under cellular conditions (e.g. in the presence of cellular thiols), it might be easier to consider the product(s) from the TTS+H₂S reaction as ‘H₂S₂ equivalents’, which could include H₂S₂, HSOH, and their downstream products RSSH. We have added this notation to the manuscript.

The chemistry of HSOH has not been well-studied, mainly due to its instability and the lack of suitable methods to produce HSOH. However, there are reports suggesting HSOH should react with H₂S to form H₂S₂ (*Free Radical Biol. Med.* **2011**, *50*, 196-205; *Chem. Rev.* **2018**, *118*, 1253-1337). It is expected that RSH should react as similarly as H₂S (e.g. attacking the S atom of HSOH). In addition, the reactivity of HSOH should be similar to sulfenic acids (RSOH). It is known that RSOH readily react with thiols (R'SH) to form disulfides (RSSR'). Based on these knowledges, HSOH should be expected to react with thiols (RSH) to produce persulfides (RSSH), not to form RSOH.

5. "The abstract is inaccurate, if not misleading, in my opinion. The system is not as simple as presented. The abstract should expose to the reader that TTS primarily creates HSOH and that there are several possibilities of how it secondarily creates S0 species in cells (including H₂S₂). TTS is an HSOH donor (not an H₂S₂ donor!), and this should be clearly communicated from the start. Selling the compound as a H₂S to H₂S₂ converter for cells is not fully appropriate in my opinion. It is also misleading to claim that localized H₂S₂ formation was achieved when in fact this experiment was only conducted in vitro with a recombinant protein (where localized production doesn't play a role). In my opinion, ascribing SQR-like activity to TTS is at least confusing, if not misleading. The activity/mechanism of SQR is very different from TTS. We can only say that TTS can compensate for certain aspects of SQR deficiency"

Based on these comments, we rewrote the abstract to clarify the points as suggested by the reviewer. Please see the new Abstract. We also clarified TTS as a potential HSOH donor throughout the manuscript. In addition, we removed the claim that TTS has 'SQR-like activity'. We just say TTS can compensate for certain aspects of SQR deficiency.

6. "The title should be changed, because it is misleading, in my opinion. There is no demonstration of a "redox-reversible booster system" in this paper. This is a speculation that can be and should be part of the discussion section, but not of the title. In my opinion, a more appropriate title would be something like this: "A compound converting H₂S to HSOH increases intracellular sulfane sulfur levels"."

Based on this comment, we decided to change the title. Our suggested title is: *2H-Thiopyran-2-Thione Sulfine (TTS), a Unique Compound For Converting H₂S to HSOH/H₂S₂ and Increasing Intracellular Sulfane Sulfur Levels.*

Reviewer 2:

"I believe that the comments/concerns from the previous reviews have been adequately addressed by the authors and the manuscript is now suitable for publication. "

We thank the reviewer for the positive comment.

Reviewer 3:

1. "The authors performed essential controls that strengthened the main conclusions of the manuscript and clarified my concerns. This work will be an exciting contribution to the community working on sulfur biochemistry. However, there are a few details that the authors may consider checking in the final version of the manuscript "

We thank the reviewer for the positive comment.

2. "HRMS spectra in Figure S5 show additional peaks that seem interesting. Close to the peak of the product TT (m/z 186.9888), a peak with slightly higher intensity is detected (m/z 188.98550), which is compatible with TT with one heavy sulfur. Additionally, the peak of 1-³⁴S (215.96038) is close to one (213.96278), compatible with compound 1 without heavy sulfurs. While the latter is expected to be formed by sulfur exchange among H₂S₂ molecules in solution, the peak of TT molecules that conserves the heavy atom is not easily predicted according to the mechanism proposed. Is it possible that this peak

appears due to natural isotope distribution? Do the authors consider that alternative reactions could be happening? Could the yield shown on page S9 (52%) be related to this observation? "

It is interesting to notice the peak of m/z 188.9855, which might be attributed to $TT\text{-}^{34}\text{S}$. In theory, the reaction between $TTS\text{-}^{34}\text{S}$ and H_2S should only provide TT, not $TT\text{-}^{34}\text{S}$. However, during the reaction, once $\text{H}_2\text{S}_2\text{-}^{34}\text{S}$ is formed, it or its degradation product ($\text{H}_2\text{S}\text{-}^{34}\text{S}$) may somehow react with $TTS\text{-}^{34}\text{S}$ to form $TT\text{-}^{34}\text{S}$. We don't feel the peak was due to natural isotope distribution. The yield (52%) was the isolated yield from this small-scale reaction and should not be related to the observation.

3. "Page 10. Lines 200-207. Here, the authors present results on H_2S_2 detection by MS after treatment of MEFs with TTS alone. To help the readers, the authors may mention that later -in Figure 7- it is shown that TTS alone can increase sulfane sulfur in MEFs. While following the thread of the text, up to this paragraph, only combinations of TTS and H_2S donors or treatment with TTS under hypoxic conditions -and not TTS alone- were able to increase sulfane levels significantly (HeLa cells in Figure 3 and NRCMs in Figure 4)."

We have moved the figure showing TTS-induced sulfur metabolome results from the Supporting Information to the main text (now Figure 5). Hopefully, that will allow the readers to more easily understand the results. We also changed the sentence to: *The results shown in Figure 5 revealed that TTS alone was able to preferentially increase the level of certain persulfides such as GSSH, H_2S_2 and H_2S_3 in MEF cells.*

4. "Figure 6-III-A. A minor observation: trace "6" is lacking in the plot. I wonder if fluorescence in trace "5" shows a slow increase due to a slow H_2S_2 formation or a slow reaction of H_2S_2 with SSP4."

Trace '6' in this figure (now Figure 7-III-A) would be the positive control using Na_2S_2 . That would give a straight line at a much higher F/F_0 than that of traces 1-5. That has been demonstrated in many of our previous papers (for example *Redox Biol.* **2022**, *57*, 102502). We didn't think trace 6 was necessary here, and adding that line would prevent us from more clearly showing the curve of traces 1-5. Nevertheless, this positive control is shown in Figure 7-III-B. To avoid any confusion, we revised the Figure legend for III-A to make sure the legend is consistent with the figure. The slow increase in trace 5 should be mainly due to the slow H_2S_2 formation as we have demonstrated in our previous work on SSP4 that the reaction between H_2S_2 with SSP4 is quite fast (within a few mins) under our experimental conditions.

5. "Figure 7-C. The new results on mitochondrial membrane potential that the authors provided are interesting and lead to fundamental questions that groups working in mitochondria biochemistry would find appealing. However, as discussed by the authors, possible causes of TTS's effect on mitochondrial membrane potential are not yet clear. Therefore, the mention of the similarity of TTS to SQR weakens a solid work since the data do not sufficiently support it: 1- the theoretical products of the reaction of TTS with H_2S -HSOH or H_2S_2 - are not produced by SQR (only downstream indirect products could be similar), 2- according to the kinetic data and the culture treatment with TT, the activity of TTS is not catalytic like SQR's, and 3- the cause of the effect of TTS on mitochondrial membrane potential is not yet clear."

This comment is similar to the one from reviewer 1. We agree that we should not compare TTS with SQR. As reviewer 1 suggested, we can just say 'TTS can compensate for certain aspects of SQR deficiency'. We have revised the manuscript accordingly.

6. "Lines 298-301. It is not clear to this reviewer why persulfides would accumulate in the mitochondria. TTS could consume H_2S in the cytosolic compartment, too. Maybe the overall decrease in H_2S is simply responsible for the effect observed."

To avoid any confusion, we have revised the sentence to “TTS could detoxify H₂S by rapidly removing H₂S from cellular media”.

In summary, we have addressed all the comments from the reviewers. We sincerely thank them as their comments have helped us improve the manuscript.

REVIEWERS' COMMENTS

Reviewer #1 (Remarks to the Author):

I am satisfied with the additional changes that have been made. I am happy to see this work published.

Reviewer #3 (Remarks to the Author):

The authors reframed the discussion well, and my main concerns were clarified. I have no further suggestions to make.